# A simplified model for transition prediction applicable to wind-turbine rotors

Thales Fava[1], Mikaela Lokatt[1], Niels Sørensen[2], Frederik Zahle[2], Ardeshir Hanifi[1], and
Dan Henningson[1]

[1]Department of Mechanics, Linné FLOW Centre, SeRC, KTH Royal Institute of Technology, Stockholm, Sweden
[2]Department of Wind Energy, Technical University of Denmark, Risø Campus, Roskilde, Denmark

**Correspondence:** Thales Fava (fava@kth.se)

**Abstract.**

This work aims to develop a simple framework for transition prediction over wind-turbine blades, including effects of the blade rotation and spanwise velocity without requiring fully three-dimensional simulations. The framework is based on a set of boundary-layer (BL) and parabolized stability equations (PSE), including rotation effects. An important element of the developed BL method is the modeling of the spanwise velocity at the boundary-layer edge. The two analyzed wind-turbine geometries correspond to a constant-airfoil and the DTU 10 MW Reference Wind Turbine blades. The BL model allows an accurate prediction of the chordwise velocity profiles. Further, for regions not too close to the stagnation point and root of the blade, profiles of the spanwise velocity agree with those from Reynolds-averaged Navier-Stokes (RANS) simulations. The model also allows predicting inflectional velocity profiles for lower radial positions, which may allow crossflow transition. Transition prediction is performed at several radial positions through an "envelope-of-envelopes" methodology. The results are compared with the $e^N$ method of Drela and Giles, implemented in the EllipSys3D RANS code. The RANS transition locations closely agree with those from the PSE analysis of a 2D mean-flow without rotation. These results also agree with those from the developed model for cases with low 3D and rotation effects, such as at higher radial positions and geometries with strong adverse pressure gradients where 2D TS waves are dominant. However, the RANS and PSE 2D models predict a later transition in the regions where 3D and rotation effects are non-negligible. The developed method, which accounts for these effects, predicted earlier transition onsets in this region (e.g., 19 % earlier than RANS at 26 % of the radius for the constant-airfoil geometry) and shows that transition may occur via highly oblique modes. These modes differ from 2D TS waves and appear in locations with inflectional spanwise velocity. However, except close to the root of the blade, crossflow transition is unlikely since the crossflow velocity is too low. At higher radial positions, where 3D and rotation effects are weaker and the adverse pressure gradient is more significant, modes with small waveangles (close to 2D) are found to be dominant. Finally, it is observed that an increase in the rotation speed modifies the spanwise velocity and increases the Coriolis and centrifugal forces, shifting the transition location closer to the leading edge. This work highlights the importance of considering the blade rotation and the three-dimensional flow generated by that in transition prediction, especially in the blade inner part.

# 1 Introduction

In wind-turbine design, accurate determination of aerodynamic loads is of importance as they are related to properties, such as performance and structural loads. Since aerodynamic loads can be influenced by the boundary-layer character, an accurate determination of the transition location can be significant to obtain a successful wind-turbine design. This has long been recognized by aerodynamicists, and significant efforts have been devoted to the development of transition models.

There are several transition models available (for a review see e.g. Saric et al., 2003; Langtry et al., 2006; Pasquale et al., 2009; Colonia et al., 2017). Some of these are based on the transport equations, such as the $\gamma$ (Colonia et al., 2017) and $\gamma - \tilde{Re}_\Theta$ equation models (Menter et al., 2006; Langtry et al., 2006; Sørensen, 2009; Menter et al., 2015; Langtry et al., 2015); other ones rely on stability analysis, such as the $e^N$ method (Smith and Gamberoni, 1956; van Ingen, 1956). These models are compatible with modern RANS solvers. In particular, the models of natural and bypass transition coupled with RANS solvers have shown good agreement with experiments on wind turbines (Özçakmak et al., 2020). The $\gamma - \tilde{Re}_\Theta$ has also been used for prediction of transition dominated by crossflow instability (Guerrero et al., 2018). More accessible measurement techniques such as ground-based thermographic imaging (Reichstein et al., 2019) have offered further data for the development, calibration, and comparison of transition models. The methods mentioned above can provide transition predictions at a relatively low computational cost, being common in engineering applications. While their accuracy has been validated for a number of two- and three-dimensional flows, further knowledge about their performance for rotating wind-turbine blades would be beneficial.

There are also more advanced transition-prediction methods, such as those based on direct numerical simulations (DNS) and parabolized stability equations (PSE) (Bertolotti et al., 1992; Simen and Dallmann, 1992), which can provide accurate transition prediction in three-dimensional flows. DNS aims at exactly resolving the flow field, and it can thus provide detailed information about velocity fluctuations within the boundary layer, based on which results about transition and turbulence characteristics can be derived. At this moment, only a few studies of the transition process on wind-turbine blades using high resolution simulations are available (Jing et al., 2020). The DNS approach for transition prediction provides accurate results, but it implies a high computational cost. With the current available computational power, simulations at Reynolds numbers corresponding to those of real wind turbines are not possible. The PSE analysis has a much lower computational cost compared to DNS (Özçakmak et al., 2020), but it provides more accurate transition predictions than the RANS approach with an algebraic-integral or transport model. However, there are limitations in the linear PSE approach, which are the inability to predict: i) transition in strongly non-parallel flows with rapid variation in the streamwise direction; ii) transition in strongly three-dimensional flows; iii) transition caused by global instability, as in the case of strong separation bubbles.

In two-dimensional flow fields, the waves causing instability are typically of the Tollmien-Schlichting (TS) type (Tollmien, 1929; Schlichting, 1933), whereas in three-dimensional flow fields, waves of crossflow type are also common (Saric et al., 2003). The former is more prone in wings with small sweep angles and very weak or adverse chordwise pressure gradients while the latter generally takes place for large sweep angles and favorable chordwise pressure gradients. Borodulin et al. (2019) showed a good agreement between linear stability results and experiments for TS waves developing over a swept wing. There were similarities between the TS waves found experimentally and those for the Blasius boundary layer, such as the shape of

the eigenfunctions and phase speed. However, the waves observed over the swept wing could propagate at a broader range of angles relative to the inviscid streamline, being more unstable at propagation angles between $25°$ and $70°$. Unlike the TS instability, crossflow instability has an inviscid origin, caused by the inflection of the crossflow velocity profile (Saric et al., 2003). Unstable crossflow modes can be triggered by noise or even microscopic surface roughness (Bippes, 1999; Gaponenko et al., 2002). The crossflow instability can manifest as stationary vortices in environments with low turbulence intensity and as travelling modes in cases with high turbulence intensity/low surface roughness. These waves can propagate at a narrower range of angles compare to TS waves and are more unstable for directions nearly perpendicular to the inviscid flow direction.

The present work aims to develop a simple model for transition prediction applicable for wind-turbine blades and to understand the effects of blade rotation on the boundary-layer flow and its stability. Firstly, a model to compute the boundary-layer profiles over the wind-turbine blades is developed. This model is based on the quasi-three-dimensional boundary-layer equations (BLE) and accounts for effects of the blade rotation and the three dimensional outer flow. A technique to obtain an approximation for the spanwise velocity is also provided, such that the only required inputs are the chordwise distribution of pressure or streamwise velocity and the blade geometry. Secondly, the $e^N$ method is employed to predict the transition locations. The $N$-factors are obtained using an existing PSE code (Hanifi et al., 1994; Hein et al., 1994) to which rotation effects are added. The developed framework is applied to two different full-scale wind-turbine geometries and the results are compared with the mean-flow and transition data from EllipSys3D RANS simulations (Michelsen, 1992, 1994; Sørensen, 1994). Transition prediction within this solver is obtained through the semiempirical $e^N$ method of Drela and Giles (Drela and Giles, 1987; Özçakmak et al., 2020). This transition model does not account for effects of the blade rotation or the three-dimensional flow. The PSE results may also indicate accuracy of the RANS prediction. Finally, effects of the rotation speed and spanwise velocity on the transition location are analyzed and the suitability of XFOIL (Drela, 1989) data as the input to the developed model is assessed.

## 2  Boundary-layer model

This section describes the boundary-layer (BL) model developed in this work.

### 2.1  Coordinate system

The coordinate system of the BL model is illustrated in Fig. 1. The blade rotates around a vertical axis at a constant angular velocity $\Omega$, and the coordinate system is fixed to the blade. Therefore, centrifugal and Coriolis forces need to be included in the fluid-dynamic equations (Kundu et al., 2016). The first coordinate direction $x_1$ follows the wing contour along a circular arc with radius $r_0$, the second coordinate direction $x_2$ is perpendicular to the $x_1$ direction in the plane tangent to the wing surface, whereas the third coordinate direction $x_3$ is defined to be in the direction normal to the surface. Hence, $x_1, x_2, x_3$ describe an orthogonal, curvilinear coordinate system. The error committed by assuming that the $x_1$ and $x_2$ directions are respectively the chordwise and spanwise directions is low. That is because the chord to radius ratio and the sweep angle are small in the analyzed wind-turbine blades. For instance, the angle between the $x_2$ and spanwise directions oscillates between $1°$ and $4°$.

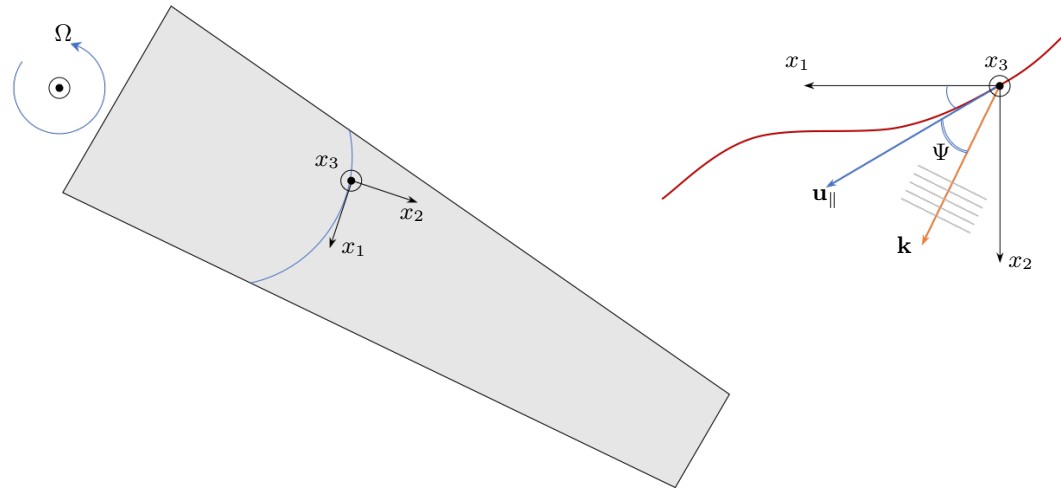

**Figure 1.** Coordinate system on the wind-turbine blade. $\Omega$ is the rotation speed, $\mathbf{u}_\parallel$ the mean-flow velocity vector projected in the $x_1 - x_2$ plane, $\mathbf{k} = (\alpha, \beta, 0)$ the wavevector and $\Psi$ the perturbation propagation angle relative to the outer streamline.

### 2.1.1 Boundary-layer equations

There are several integral formulations of the boundary-layer equations (BLE) (Du and Selig, 2000; Dumitrescu and Cardos, 2011; Drela, 2013; Garcia et al., 2014). However, a differential formulation is expected to be more accurate than its integral counterpart because the latter requires closure relations which are found through empirical relations (van Garrel, 2004). For this reason, a differential formulation is selected in the present case. When expressed in the coordinate system described in Sect. (2.1), the differential form of the BLE can be written as (Warsi, 1999)

$$\frac{\partial}{\partial x_1}(\rho h_2 h_3 u_1) + \frac{\partial}{\partial x_2}(\rho h_1 h_3 u_2) + \frac{\partial}{\partial x_3}(\rho h_1 h_2 u_3) = 0, \tag{1}$$

$$\rho\left(\frac{u_1}{h_1}\frac{\partial u_1}{\partial x_1} + \frac{u_2}{h_2}\frac{\partial u_1}{\partial x_2} + \frac{u_3}{h_3}\frac{\partial u_1}{\partial x_3} + \frac{1}{h_1 h_2}\left(\frac{\partial h_1}{\partial x_2}u_1 u_2 - \frac{\partial h_2}{\partial x_1}u_2^2\right)\right) =$$
$$-\frac{1}{h_1}\frac{\partial p}{\partial x_1} + \frac{1}{Re}\frac{1}{h_3}\frac{\partial}{\partial x_3}\left(\frac{\mu}{h_3}\frac{\partial u_1}{\partial x_3}\right) + \rho\left(2\Omega_3 u_2 + \frac{\Omega^2}{2h_1}\frac{\partial r^2}{\partial x_1}\right), \tag{2}$$

$$\rho\left(\frac{u_1}{h_1}\frac{\partial u_2}{\partial x_1} + \frac{u_2}{h_2}\frac{\partial u_2}{\partial x_2} + \frac{u_3}{h_3}\frac{\partial u_2}{\partial x_3} + \frac{1}{h_1 h_2}\left(\frac{\partial h_2}{\partial x_1}u_1 u_2 - \frac{\partial h_1}{\partial x_2}u_1^2\right)\right) =$$
$$-\frac{1}{h_2}\frac{\partial p}{\partial x_2} + \frac{1}{Re}\frac{1}{h_3}\frac{\partial}{\partial x_3}\left(\frac{\mu}{h_3}\frac{\partial u_2}{\partial x_3}\right) + \rho\left(-2\Omega_3 u_1 + \frac{\Omega^2}{2h_2}\frac{\partial r^2}{\partial x_2}\right), \tag{3}$$

$$\rho c_p\left(\frac{u_1}{h_1}\frac{\partial T}{\partial x_1} + \frac{u_2}{h_2}\frac{\partial T}{\partial x_2} + \frac{u_3}{h_3}\frac{\partial T}{\partial x_3}\right) = \frac{1}{RePr}\frac{1}{h_3}\frac{\partial}{\partial x_3}\left(\frac{\kappa}{h_3}\frac{\partial T}{\partial x_3}\right) +$$
$$(\overline{\gamma} - 1)M^2\left\{\frac{u_1}{h_1}\frac{\partial p}{\partial x_1} + \frac{u_2}{h_2}\frac{\partial p}{\partial x_2} + \frac{\mu}{Re}\left[\left(\frac{\partial u_1}{\partial x_3}\right)^2 + \left(\frac{\partial u_2}{\partial x_3}\right)^2\right]\right\}. \tag{4}$$

In these equations, $c_p, \overline{\gamma}, \kappa, \mu, M, Re$, and $Pr$ denote specific heat capacity at constant pressure, ratio of specific heats, thermal conductivity, dynamic viscosity, Mach number, Reynolds number based on a reference length $l_0$, and Prandtl number, respectively. Moreover, $\rho, p$, and $T$ denote density, pressure, and temperature, whereas $\mathbf{u}$ and $\mathbf{\Omega}$ represent velocity and rotation, respectively. $h_i$ are the Lamé coefficients, where $h_i^2 = g_{ii}$ and $g_{ij}$ is the metric tensor. Note that since the coordinate system is orthogonal $g_{ij} = 0$ for $j \neq i$. The subscripts 1, 2, and 3 indicate components in the respective $x_1, x_2$, and $x_3$ directions. $r$ is the radial position.

In the BL model, the chordwise curvature of the wing model is neglected, while the radial curvature is considered. Thus, the metric vector becomes

$$h_1 = \frac{x_2 + r_0}{r_0}, \quad h_2 = 1, \quad h_3 = 1. \tag{5}$$

Since the code is intended for analysis of laminar flows, turbulent fluctuations and statistics need not be considered. In order to obtain a well-conditioned system which solution is compatible with the subsequent PSE analysis, the terms in the system of Eqs. (1) to (4) are normalized by the reference quantities given in Table 1. The value of $l_0$ is set to $c_0$, the chord of the airfoil at the radial position $r_0$, where the analysis is performed.

**Table 1.** Reference values. $\infty$ denotes freestream values.

| Variable | Reference value |
| --- | --- |
| Length | $l_0$ |
| Velocity | $u_\infty$ |
| Angular velocity | $u_\infty/l_0$ |
| Density | $\rho_\infty$ |
| Pressure | $p_\infty$ |
| Temperature | $T_\infty$ |
| Dynamic viscosity | $\mu_\infty$ |
| Thermal conductivity | $\kappa_\infty$ |

### 2.1.2 Approximations of the spanwise derivatives

As they stand, the BL equations are dependent on all three coordinate directions so that their numerical solution requires a full volume discretization. A three-dimensional discretization can result in a solution procedure that is costly in terms of computational capacity and CPU time. By employing approximate models for the derivative terms in the $x_2$ direction, instead of exact expressions, one can obtain a quasi-three-dimensional model requiring discretization in the $x_1$ and $x_3$ directions only. The reduced dimension of the discretization typically results in significant savings in computational cost and meshing effort. Furthermore, a judicious selection of the model for the $x_2$ derivative can provide accurate mean-flows. These beneficial properties lead a quasi-three-dimensional model to be employed in the present work.

Similarity solutions for rotating flows suggest that the velocity in the $x_1$ direction can be assumed to depend on the $x_2$ coordinate linearly (Greenspan, 1968; Hernandez, 2011). This approximation is employed in the present work, together with the further assumption that the velocity in the $x_2$ direction, pressure, and temperature does not depend on $x_2$. Thus,

$$u_1 = u_{1_0} \frac{x_2 + r_0}{r_0}, \quad u_2 = u_{2_0}, \quad p = p_0, \quad T = T_0. \tag{6}$$

The subscript 0 denotes evaluation at the radial location $r_0$. This choice can result in a momentum imbalance in the $x_2$ direction at the boundary-layer edge, as pointed by Sturdza (2003) for swept-wing flows. Sturdza argued that the imbalance could be compensated by defining an additional source term $A$ that accounts for the momentum difference. The extra source term is then multiplied by a blending function $f(x_3)$ and added to the right-hand side of the spanwise momentum equation (Eq. (3)). $A$ is found by considering momentum balance at the boundary-layer edge. With the current approximation of spanwise derivatives and curvature terms, $A$ becomes

$$A = \rho u_{1_e} \frac{\partial u_{2_e}}{\partial x_1} - \frac{\rho u_{1_e}^2}{r_0} - \rho\left(-2\Omega_3 u_1 + \Omega^2 r_0\right), \tag{7}$$

where the subscript $e$ denotes evaluation at the boundary-layer edge. The blending function is selected to linearly depend on the wall-normal distance inside the boundary layer, i.e.,

$$f(x_3) = \frac{x_3}{x_{3_e}}. \tag{8}$$

### 2.1.3 Discretization of BLE

The spanwise approximations described in Sect. (2.1.2) make the system of the BLE (Eqs. (1) to (4)) include only derivatives in the $x_1$ and $x_3$ directions. The derivatives in the $x_3$ direction are evaluated using a second-order central finite-difference scheme, whereas the derivatives in the $x_1$ direction are evaluated using a second-order backward Euler finite-difference scheme. The BLE can be expressed as

$$\mathbf{A_1}\mathbf{\Phi} + \mathbf{A_2}\frac{\partial\mathbf{\Phi}}{\partial x_3} + \mathbf{A_3}\frac{\partial^2\mathbf{\Phi}}{\partial x_3^2} + \mathbf{A_4}\frac{\partial\mathbf{\Phi}}{\partial x_1} = \mathbf{A_5}, \tag{9}$$

where $\mathbf{\Phi} = (u_1, u_2, T)^T$ denotes the vector of primary variables. The density is calculated from the temperature and pressure using the equation of state and the BL approximation of pressure being constant inside the boundary layer. The components of the matrices $\mathbf{A_1}, \mathbf{A_2}, \mathbf{A_3}, \mathbf{A_4}$, and $\mathbf{A_5}$ are found by collecting terms in Eqs. (1) to (4).

The solution is computed by space marching in the $x_1$ direction. Uniform boundary conditions are assumed at the inflow. The attachment-line equations (Cebeci, 1999) are solved at the first inflow node, since the BLE are ill-conditioned when $u_1$ is equal to zero. Because of the boundary-layer singularity (Goldstein, 1948), the system of equations can become strongly ill-conditioned if flow separation is encountered. However, the present code is intended to be used for transition prediction, and separation within a laminar-flow region typically causes transition. Therefore, the separation point can be taken as a reasonable approximation of the transition location, and the issue is circumvented.

## 2.2 Edge velocity model

The velocity in the $x_2$ direction at the boundary-layer edge is required as input to the quasi-three-dimensional BL model. In order to avoid the necessity of a costly simulation to obtain it, a model for $u_{2_e}$ is devised with inspiration from the conical-wing approximation (Cebeci, 1999; Sturdza, 2003) . An approximation for $u_{2_e}$ is obtained by combining the Euler equation in the $x_2$ direction with an approximation for the variation of the pressure coefficient in this direction. The Euler equation in the $x_2$ direction can be written as (Warsi, 1999)

$$\rho \left[ \frac{u_1}{h_1} \frac{\partial u_2}{\partial x_1} + \frac{u_2}{h_2} \frac{\partial u_2}{\partial x_2} + \frac{u_3}{h_3} \frac{\partial u_2}{\partial x_3} + \frac{1}{h_1 h_2} \left( \frac{\partial h_2}{\partial x_1} u_1 u_2 - \frac{\partial h_1}{\partial x_2} u_1^2 \right) + \frac{1}{h_2 h_3} \left( \frac{\partial h_2}{\partial x_3} u_2 u_3 - \frac{\partial h_3}{\partial x_2} u_3^2 \right) \right] = -\frac{1}{h_2} \frac{\partial p}{\partial x_2} + F_{rot_2}, \quad (10)$$

where

$$F_{rot_2} = \rho \left[ 2 u_3 \Omega_1 - 2 u_1 \Omega_3 - \left( \Omega_2 x_3 - \Omega_3 x_2 \right) \Omega_3 + \left( \Omega_1 x_2 - \Omega_2 x_1 \right) \Omega_1 \right]. \quad (11)$$

We assume that $\dfrac{u_2}{h_2} \dfrac{\partial u_2}{\partial x_2} \approx 0$, based on the fact that the flow and the variations in the $x_2$ direction have a small magnitude. A second hypothesis is that $\dfrac{u_3}{h_3} \dfrac{\partial u_2}{\partial x_3} \approx 0$, built on the evidence that the flow and variations in the normal direction at the boundary-layer edge are small. Since $u_3 \approx 0$ and $\Omega_1 \approx 0$, the term $2 u_3 \Omega_1$ is neglected in Eq. (11). However, the terms $\dfrac{u_3}{h_3} \dfrac{\partial u_2}{\partial x_3}$ and $2 u_3 \Omega_1$ may be relevant close to the stagnation point because $u_3 \approx ||\mathbf{u}||$ and $\Omega_1 \approx ||\mathbf{\Omega}||$. Therefore, Eq. (10) should be valid only after a slightly downstream distance from the stagnation point. Moving all terms except the one containing $\dfrac{\partial u_2}{\partial x_1}$ to the right-hand side, dividing both sides of the equation by $\rho \dfrac{u_1}{h_1}$, and including the scale factors given by Eq. (5) yield

$$\frac{\partial u_2}{\partial x_1} = \frac{h_1}{\rho u_1} \left( -\frac{\partial p}{\partial x_2} + F_{rot_2} + \rho u_1^2 \frac{\partial h_1}{\partial x_2} \right). \quad (12)$$

All terms on the right-hand side are known except for the $x_2$ pressure gradient. An approximation for this term can be found by rewriting the definition of the pressure coefficient with the reference speed equals to the rotational one, i.e.,

$$p = C_p \frac{1}{2} \rho \left( \Omega r_0 \right)^2 + p_\infty, \quad (13)$$

and assuming that

$$C_p = C_{p_0} \frac{r^2}{r_0^2} \frac{\alpha}{\alpha_0}, \quad (14)$$

where $C_{p_0}$ is the pressure coefficient at the radial position $r_0$ and $r = x_2 + r_0$. Equation (14) models the variation in $C_p$ due to the change of the reference velocity with $r$, as well as a first-order variation in $C_p$ due to the change of the angle of attack $\alpha$. The latter is defined as

$$\alpha = \tan^{-1} \left( \frac{w_\infty}{\Omega r_0} \right) + \theta \left( x_2 \right), \quad (15)$$

with $w_\infty$ and $\theta$ representing the incoming-flow velocity and the geometric twist angle, respectively. Note that Eq. (14) is singular for $\alpha_0 = 0$ and may not be very accurate for small values of $\alpha_0$. Therefore, some other approximations may be more

suitable for these cases. With inspiration from the conical-wing approximation (Cebeci, 1999; Sturdza, 2003), $C_{p_0}$ is assumed to be constant along conical lines. These lines as well as other parameters related to the conical-wing approximation are illustrated in Fig. 2.

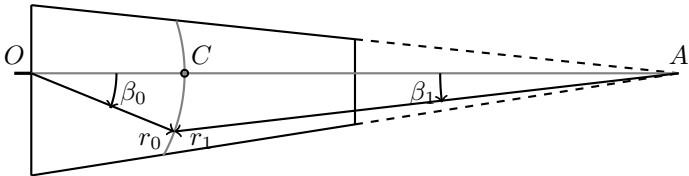

**Figure 2.** Conical parameters. $O$ and $A$ are the center of rotation and the cone apex, respectively. Lines of constant $\beta_1$ are the conical lines.

With this assumption, the derivative of $C_{p_0}$ in the $x_2$ direction can be related to its derivative in the $x_1$ direction by

$$\frac{\partial C_{p_0}}{\partial x_2} = -\tan\left(\beta_1 + \beta_0\right)\frac{\partial C_{p_0}}{\partial x_1}. \tag{16}$$

The angles $\beta_1$ and $\beta_0$ are defined as

$$\beta_1 = \sin^{-1}\left(\frac{x_{1_c} - x_1}{r_1}\right), \quad \beta_0 = \sin^{-1}\left(\frac{x_{1_c} - x_1}{r_0}\right), \tag{17}$$

where $x_{1_c}$ denotes the $x_1$ coordinate of point $C$, where the line connecting the center of rotation $O$ and the cone apex $A$ intersects the arc with radius $r_0$. These assumptions lead to an expression for the pressure derivative, given by

$$\frac{\partial C_p}{\partial x_2} = -\tan\left(\beta_1 + \beta_0\right)\frac{\partial C_{p_0}}{\partial x_1}\frac{r^2}{r_0^2}\frac{\alpha}{\alpha_0} + C_{p_0}\left(\frac{\alpha}{\alpha_0}\frac{2r}{r_0^2} + \frac{r^2}{r_0^2}\frac{1}{\alpha_0}\frac{\partial \alpha}{\partial x_2}\right). \tag{18}$$

Inserting Eqs. (13), (14), and (18) in Eq. (12) provides an expression that can be integrated along $x_1$ to obtain the distribution of $u_{2_e}$ in this direction. However, it is necessary to obtain an approximation for $u_{2_e}$ at the initial point of integration. In order to do that, we use as inspiration the swept-wing approximation (Cebeci, 1999) and assume that $u_{2_e}$ can be approximated by the velocity over a conical line (see Fig. 2). This approximation yields

$$u_{2_e} = (2\Omega r_0 - u_{1_e})\tan\left(\beta_1 + \beta_0\right), \tag{19}$$

where $2\Omega r_0$ is a reference velocity. However, Eq. (19) is not very accurate if $u_{1_e}$ is small, as is the case near the attachment line. Thus, it is advisable to start the integration at a position $x_{1_0}$ downstream of the attachment line, where $u_{1_e}$ has a value that is comparable to the freestream velocity. An approximate initial value for $u_{2_e}$ at $x_{1_0}$ can be found from

$$u_{2_e}(x_{1_0}) = [2\Omega r_0 - u_{1_e}(x_{1_0})]\left(x_{1_c} - x_{1_0}\right)\frac{r_0 + r_1}{r_0 r_1}. \tag{20}$$

## 3   PSE

The coordinate system employed in the PSE analysis is the one in Fig. 1. The PSE is derived from the continuity, momentum, energy, and state equations (Hanifi et al., 1994; Kundu et al., 2016), as shown in Eqs. (21) to (24). Because of the complexity

of performing a full three-dimensional analysis, periodicity is assumed in the $x_2$ direction. Moreover, rotation terms are added to the momentum equations.

$$\frac{\partial \rho}{\partial t} + \nabla \cdot (\rho \mathbf{u}) = 0, \tag{21}$$

$$\rho \left[ \frac{\partial \mathbf{u}}{\partial t} + (\mathbf{u} \cdot \nabla) \mathbf{u} \right] = -\nabla p + \frac{1}{Re} \nabla \left[ \lambda (\nabla \cdot \mathbf{u}) \right] + \frac{1}{Re} \nabla \cdot \left[ \mu \left( \nabla \mathbf{u} + \nabla \mathbf{u}^T \right) \right] + \mathbf{F_{rot}}, \tag{22}$$

$$\rho c_p \left[ \frac{\partial T}{\partial t} + (\mathbf{u} \cdot \nabla) T \right] = \frac{1}{RePr} \nabla \cdot (\kappa \nabla T) + (\overline{\gamma} - 1) M^2 \left[ \frac{\partial p}{\partial t} + (\mathbf{u} \cdot \nabla) p + \frac{1}{Re} \Phi \right], \tag{23}$$

$$\overline{\gamma} M^2 p = \rho T, \tag{24}$$

$$\mathbf{F_{rot}} = -\rho \left[ 2\mathbf{\Omega} \times \mathbf{u} + \mathbf{\Omega} \times (\mathbf{\Omega} \times \mathbf{x}) \right], \tag{25}$$

$$\Phi = \lambda (\nabla \cdot \mathbf{u})^2 + \frac{1}{2} \mu \left( \nabla \mathbf{u} + \nabla \mathbf{u}^T \right)^2, \tag{26}$$

where $\lambda = -\frac{2}{3}\mu$ denotes the second viscosity coefficient under the Stokes hypothesis. The quantities in these equations have been normalized with the reference values given in Table 1.

The flow can be decomposed as

$$\mathbf{q}(x_1, x_3, t) = \overline{\mathbf{q}}(x_1, x_3) + \epsilon \, \tilde{\mathbf{q}}(x_1, x_3, t), \tag{27}$$

where $t$ denotes time, $\mathbf{q} = (u_1, u_2, u_3, T, \rho)^T$. Here, pressure is eliminated using the equation of state. The bar denotes the mean-flow variables from the BL model or RANS, tilde, the perturbation quantities, and $\epsilon \ll 1$ (Hanifi et al., 1994; Hein et al., 1994) . The perturbation part has the form

$$\tilde{\mathbf{q}}(x_1, x_3, t) = \hat{\mathbf{q}}(x_1, x_3) e^{i\Theta}, \tag{28}$$

where $\hat{\mathbf{q}}(x_1, x_3)$ denotes the slowly varying part of the perturbation, $i$ the imaginary unit, and $\Theta$ is

$$\Theta = \int_{x_0}^{x_1} \alpha(x') \, dx' + \beta x_2 - \omega t, \tag{29}$$

where $\alpha$ and $\beta$ are the wavenumber in the $x_1$ and $x_2$ directions, respectively, whereas $\omega$ denotes the temporal angular frequency of the disturbance. $x_0$ is the chordwise coordinate of the initial point of analysis. Including these relations in Eqs. (21) to (24), assuming that the variation in the $x_1$ direction is weak compared to the variation in the $x_3$ one (there is a scale of $1/Re$ between them), neglecting terms of order $\epsilon^2$, and collecting the terms we obtain a system of the form

$$\mathbf{B_1} \hat{\mathbf{q}} + \mathbf{B_2} \frac{\partial \hat{\mathbf{q}}}{\partial x_3} + \mathbf{B_3} \frac{\partial^2 \hat{\mathbf{q}}}{\partial x_3^2} + \mathbf{B_4} \frac{\partial \hat{\mathbf{q}}}{\partial x_1} = \mathbf{0}. \tag{30}$$

In addition, the following normalization condition is used

$$\int_0^\infty \hat{\mathbf{q}}^* \frac{\partial \hat{\mathbf{q}}}{\partial x_3} dx_3 = 0, \tag{31}$$

where the superscript $*$ denotes the complex conjugate (Hanifi et al., 1994). The following boundary conditions are employed

$$
\begin{cases}
\hat{u}_1 = \hat{u}_2 = \hat{u}_3 = \hat{T} = 0, & \text{for } x_3 = 0, \\
\hat{u}_1, \hat{u}_2, \hat{u}_3, \hat{T} \to 0, & \text{for } x_3 \to \infty.
\end{cases}
\tag{32}
$$

Notice that the far-field condition $\hat{u}_3 \to 0$ can be replaced by $\hat{\rho} \to 0$. The derivatives in the $x_3$ direction are computed with a fourth-order compact finite-difference scheme, whereas the derivatives in the $x_1$ direction are computed with a second-order compact finite-difference scheme. Given initial values of $\alpha$ and $\beta$, the growth of the disturbances along $x_1$ is evaluated by

marching Eq. (30) in the $x_1$ direction.

In the $e^N$ method, transition location is predicted based on the amplification of disturbances presented by the $N$-factors computed as

$$
N = \ln\left(A/A_0\right) = \int_{x_I}^{x} \sigma(x')dx',
\tag{33}
$$

where $A$ is the amplitude of the perturbations ($A_0 = A(x_0)$), $x_I$ the location where the perturbation first start to grow and $\sigma$

the growth rate of the perturbation kinetic energy $E$ defined as (Hanifi et al., 1994)

$$
\sigma = \frac{1}{h_1}\left[-\text{Im}(\alpha) + \text{Re}\left(\frac{1}{E}\frac{\partial E}{\partial x_1}\right)\right], \quad E = \int_0^\infty \overline{\rho}\left(\hat{u}_1^2 + \hat{u}_2^2 + \hat{u}_3^2\right) dx_3.
\tag{34}
$$

Here, consistent with the PSE framework, we use an "*envelope-of-envelopes*" approach meaning that transition is predicted based on the envelope of the amplification curves computed for fixed values of $\omega$ and $\beta$ (see e.g. Arnal and Casalis, 2000).

## 4    Results

The results of the proposed approach are compared to those from the EllipSys3D RANS code. This solver is based on the incompressible Navier-Stokes equations and employs a block-structured, finite-volume discretization, including a second-order upwind scheme for the discretization of convective terms and a central difference scheme for the discretization of the viscous ones. Turbulence is modeled using the SST $k-\omega$ turbulence model (Menter, 1993) and the transition prediction is performed using an $e^N$ method (Drela and Giles, 1987) combined with a model for the turbulence intermittency factor $\gamma$ (Özçakmak et al.,

2020). The intermittecncy function is defined as

$$
\gamma = 1 - \exp\left\{-(x - x_{tr})^2 \left(\frac{U_{e,tr}}{\nu}\right)^2 \hat{n}\sigma\right\}, \text{ for } x \geq x_{tr},
\tag{35}
$$

where $x$ is the chordwise position (measured from the stagnation line, $x_{tr}$ is the chordwise position of the transition onset, $\nu$ is the kinematic viscosity, $\sigma$ is the spot propagation rate, $\hat{n}$ is the nondimensional spot formation rate, and $U_{e,tr}$ is the edge velocity at the chordwise position of the transition onset (Mayle, 1999). For laminar flow, i.e., $x < x_{tr}$, $\gamma = 0$, and for fully

turbulent flow, $\gamma = 1$.

## 4.1 Test cases

Two different full-scale wind-turbine rotors are investigated. Both have three blades, and their geometries are illustrated in Fig. 3. The shaded colors show a normalized measure of the axial position of each mesh point on the blade surface. The first geometry (Geometry 1) has a tapered and twisted blade with a symmetric NACA 63-018 airfoil profile along its entire span. It was mainly designed to allow the investigation of the accuracy of the conical-wing-based edge velocity model when applied to a geometry respecting its geometrical assumptions. The second geometry (Geometry 2) corresponds to the blade of the DTU 10 MW Reference Wind Turbine (Bak et al., 2012). It has a tapered and twisted blade with spanwise-varying cross-sectional properties. This enables the evaluation of our quasi-three-dimensional model when applied to a general wind-turbine blade geometry. It is assumed that the flows over the three blades are similar so that it is sufficient to analyze one blade. We focus on the suction side of the blade since transition often occurs earlier there. Attachment-line transition is not expected to occur as the attachment-line Reynolds number $\overline{R} = 41$ and 15 for Geometries 1 and 2, respectively, where $\overline{R} = (u_\infty R_{le} \sin\phi \tan\phi/(2\nu))^{1/2}$, $u_\infty$ is the incoming infinite velocity, $R_{le}$ is the curvature radius of the leading edge, and $\phi$ is the sweep angle. This is well below the threshold of 250 for contamination (Poll, 1978).

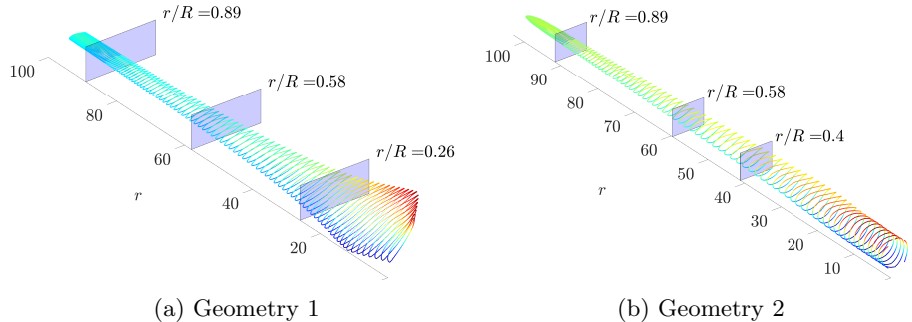

(a) Geometry 1          (b) Geometry 2

**Figure 3.** Wind-turbine blades with radial sections of analysis. The surface is colored with a normalized measure of the axial position of the mesh point. The radial coordinate $r$ is given in meters. $R$ is the radius of the wind-turbine rotor.

The main parameters of the two cases are given in Table 2. Both were computed using a temperature of 287.5 K, density of $1.225\,\mathrm{kg} \cdot \mathrm{m}^{-3}$, dynamic viscosity of $1.784 \cdot 10^{-5}\,\mathrm{kg} \cdot \mathrm{m}^{-1} \cdot s^{-1}$, ratio of specific heats of 1.4, and gas constant of $287\,\mathrm{J} \cdot \mathrm{kg}^{-1} \cdot \mathrm{K}^{-1}$. The meshes used for the RANS computations of Geometries 1 and 2 have $15.5 \cdot 10^6$ nodes, of which $118 \cdot 10^3$ are surface ones. The boundary layer is discretized with approximately 50 nodes in the wall-normal direction. The corresponding meshes for the BL and PSE models have 200 and 500 points in this direction, respectively. This level of discretization provided spatially converged results for test cases. However, a lower number of grid points could be used for increased performance when computing the envelope of $N$-factors with the PSE.

For the easiness of the reader, the acronyms of the methods used in the following sections are summarized in Table 3.

**Table 2.** Physical parameters of the wind turbines.

|  | Geometry 1 | Geometry 2 |
|---|---|---|
| Number of blades | 3 | 3 |
| Radius [m] | 100.0 | 102.9 |
| Position of maximum chord [m] | 12.0 | 30.0 |
| Root chord [m] | 7.5 | 5.4 |
| Tip chord [m] | 3.7 | 2.9 |
| Maximum chord [m] | 14.2 | 6.0 |
| Root twist angle [°] | $-90.0$ | 0.0 |
| Tip twist angle [°] | 0.0 | -4.0 |
| Twist angle at position of maximum chord [°] | $-17.0$ | -11.3 |
| Blade cross section (airfoil profile) | NACA 63-018 | FFA-W3-241 with decreasing thickness up to 2/3 of the radius |
| Rotational velocity [$\mathrm{rad \cdot s^{-1}}$] | 0.64 | 0.90 |
| Horizontal free stream velocity [$\mathrm{m \cdot s^{-1}}$] | 8.0 | 10.0 |
| Tip-speed ratio | 8.0 | 9.3 |
| Average chord Reynolds number | $1.48 \cdot 10^7$ | $1.55 \cdot 10^7$ |

**Table 3.** Acronyms of the employed methods.

| Acronym | Description |
|---|---|
| RANS | Results from RANS simulations performed with the EllipSys3D code |
| EVMR | Edge Velocity Model with $u_{1_e}(x_1)$ from RANS |
| EVMX | Edge Velocity Model with $u_{1_e}(x_1)$ from XFOIL |
| BLR | Boundary Layer Model with $u_{1_e}(x_1)$ from RANS and $u_{2_e}(x_1)$ from EVMR |
| BLX | Boundary Layer Model with $u_{1_e}(x_1)$ from XFOIL and $u_{2_e}(x_1)$ from EVMX |
| BLR 2D | 2D boundary layer equations (no rotation) with $u_{1_e}(x_1)$ from RANS |
| RANS ($\gamma = 0.01$) | Transition locations obtained from RANS for an intermittency factor $\gamma = 0.01$ |
| PSER | Transition locations obtained from PSE for BLR velocity profiles |
| PSEX | Transition locations obtained from PSE for BLX velocity profiles |
| PSER 2D | Transition locations obtained from PSE (no rotation) for BLR 2D velocity profiles |

## 4.2  Pressure distributions

The pressure distributions from RANS and XFOIL are shown in Fig. 4. Close agreement is obtained for the middle and outer radial locations of Geometry 1. For Geometry 2 and the inner radial location of Geometry 1, XFOIL results indicate a less

severe pressure drop along the airfoil, although RANS and XFOIL pressure gradients are close to each other for the initial chordwise extent of the airfoils. For Geometry 1 at $r_0/R = 0.26$ and Geometry 2 at $r_0/R = 0.89$, XFOIL results also indicate small separation bubbles at $x_1 \approx 0.45$, which are not present in RANS distributions. A possible source of those differences is the mismatch between the angles of attack (AoA) of XFOIL and RANS. The XFOIL computations are for an AoA calculated based on the inflow velocity and that generated by the blade rotation, which may differ from the actual AoA in the RANS simulation. Moreover, XFOIL $C_p$ distributions were obtained for a two-dimensional section of the wing, without considering its spanwise variation and the three-dimensionality of the flow present in the RANS results. Those effects are particularly important for Geometry 1 at $r_0/R = 0.26$.

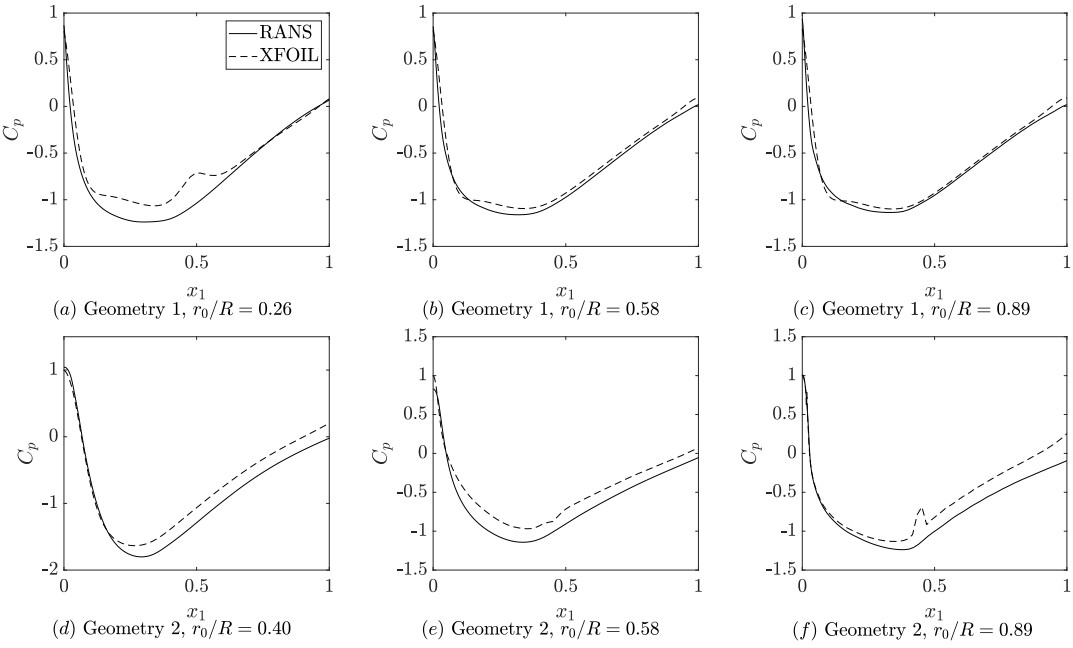

**Figure 4.** Comparison between XFOIL and RANS pressure distributions for the suction side of the airfoils of Geometries 1 and 2 at three radial positions.

### 4.3 Spanwise edge velocity

Here, we compare the chordwise distributions of spanwise velocity at the edge of the boundary layer $u_{2_e}$ obtained with RANS simulations and the edge velocity model (EVM). The analyses are performed at three radial locations $r_0$ in the inner ($r_0/R = 0.26$ and $0.40$), middle ($r_0/R = 0.58$), and outer ($r_0/R = 0.89$) parts of the blade, where $R$ is the radius of the rotor. The inner section for Geometry 2 ($r_0/R = 0.40$) is chosen after the location of the maximum chord at $r_0/R = 0.30$.

Figures 5a, 5c, and 5e present the results for Geometry 1. The spanwise velocity is of the order of 1 % of the freestream velocity, except close to the stagnation point, where it can reach higher values. EVMR and RANS results agree for the middle and outer radial locations after 10 % of the chord. The differences between EVMX and RANS results are also small for these

locations. The small overestimation of $u_{2_e}$ of the EVMX method compared to RANS/EVMR is related to the smaller flow acceleration predicted by XFOIL compared to its RANS counterpart (see Eq. (12)). The differences between the EVM and RANS results are larger at the inner radial position and close to the stagnation point. The reason is that the approximation for the spanwise pressure gradient given by Eq. (16) is more accurate at large radii and chordwise positions. This approximation relies on the assumption of $C_{p_0}$ being constant over conical lines, which may not be respected at the mentioned locations due to the strong variation of the geometry in the radial direction and the flow three-dimensionality.

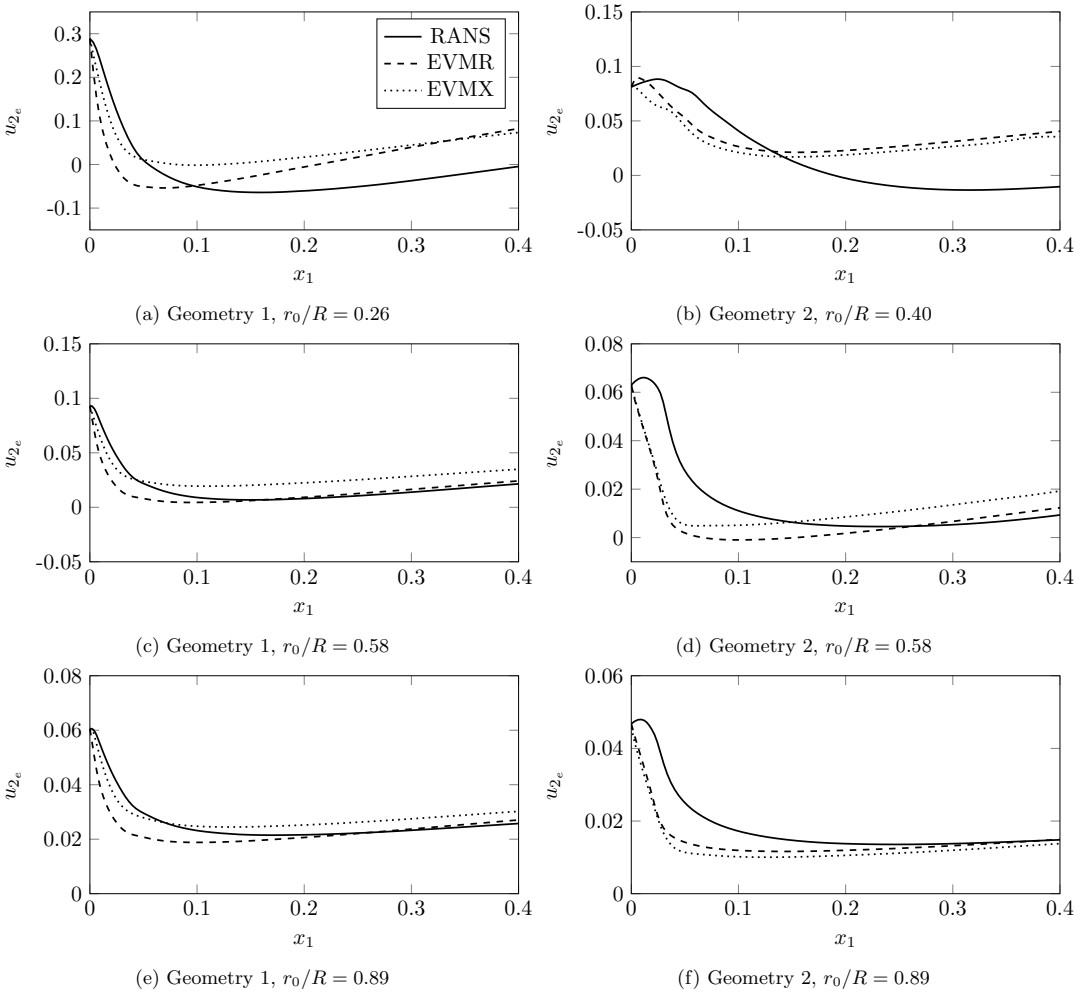

(a) Geometry 1, $r_0/R = 0.26$

(b) Geometry 2, $r_0/R = 0.40$

(c) Geometry 1, $r_0/R = 0.58$

(d) Geometry 2, $r_0/R = 0.58$

(e) Geometry 1, $r_0/R = 0.89$

(f) Geometry 2, $r_0/R = 0.89$

**Figure 5.** Spanwise edge velocity.

The results for Geometry 2 are presented in Figs. 5b, 5d, and 5f. At the inner radial location, $r_0/R = 0.40$, EVMR and EVMX results indicate a higher spanwise velocity than RANS, similarly to Geometry 1. In previous analysis of Geometry 2 (Zahle et al., 2014), a region of three-dimensional flow radially pumped from the root to $r_0/R = 0.36$ was observed. Moreover, a separation bubble is also present from the root to almost $r_0/R = 0.40$ (Horcas et al., 2017). These factors increase the flow

three-dimensionality at the inner radial part of the blade, making it more difficult for the quasi-three-dimensional BL model to capture the flow features correctly. However, the agreement between EVM and RANS results improves with $r_0/R$ and $x_1$. This is particularly true at $r_0/R = 0.58$ and $0.89$ after $15$ % of the chord. The differences between EVM and RANS velocity distributions were expected to be higher for Geometry 2 because the spanwise variation of the airfoil spurs changes in the $C_p$ along conical lines. The higher spanwise velocity of Geometry 1, especially at the inner radial location, associated with inflectional spanwise velocity profiles, as shown in the next section, indicates a higher potential for crossflow instability. These results suggest that the edge velocity model can provide a reliable approximation for $u_{2_e}$ for radial positions not too close to the root of the blade and stagnation point. The results are expected to be more accurate for geometries respecting the assumptions of the model and generating a less three-dimensional flow, such as Geometry 1.

## 4.4 Velocity profiles

We present the chordwise and spanwise velocity profiles obtained with RANS simulations and the boundary-layer model as a function of the normal coordinate $x_3$ nondimensionalized by the BL thickness $\delta$. Two chordwise positions are analyzed for each radial location. Figure 6 presents the results for Geometry 1. The BLR, BLX, and BLR 2D profiles of chordwise velocity are in close agreement with the RANS results for all locations. They resemble the Falkner-Skan type of profiles for an accelerating flow and seem to be little affected by three-dimensionality since they agree with the BLR 2D solution. Further downstream, around $x_1 = 0.40$, the flow starts to decelerate (see Fig. 4), which may allow the appearance of a viscous instability of the Tollmien-Schlichting (TS) type. These conclusions also apply to Geometry 2, whose results are shown in Fig. 7. The qualitative behavior of the chordwise velocity profiles is similar. However, the flow starts to decelerate earlier, at around $x_1 = 0.30$, for the inner radial position and approximately $x_1 = 0.40$ for the middle and outer radial locations. Therefore, an earlier transition may be expected for Geometry 2 at $r_0/R = 0.40$.

The spanwise velocity at the inner radial position of Geometry 1 is directed towards the root of the blade as portrayed in Figs. 6a and 6b. This reverse flow supports the hypothesis of considerable three-dimensionality at radial locations closer to the root of the blade (Du and Selig, 2000). Although the BLR and BLX profiles of spanwise velocity are close to each other, they indicate a positive velocity (flow towards the tip of the blade) whereas the spanwise velocity profile from RANS is only positive in the near-wall region. The RANS, BLR, and BLX spanwise velocity profiles present inflection points. Therefore, they are susceptible to an inviscid instability of the crossflow type. Other cases with inflection of the spanwise velocity profile are the RANS and BLR results at $r_0/R = 0.58$ of Geometries 1 and 2 (Figs. 6d, 7c, and 7d), and the RANS results at $r_0/R = 0.40$ of Geometry 2 (Fig. 7b).

The BLR and RANS spanwise velocity profiles are in close agreement at the middle and outer radial positions of Geometry 1 as presented in Figs. 6c, 6d, 6e, and 6f. The higher values obtained with the BLX approach in those cases are caused by the larger $u_{2_e}$ predicted with the edge velocity model (EVMX). The same occurs at the outer radial location of Geometry 2, as shown in Figs. 7e and 7f, in which BLR and RANS spanwise velocity profiles agree, but the result from BLX overestimates $u_{2_e}$. Nonetheless, the shapes of the BLX profiles agree with that of the other methods, indicating that the mismatch is only due to the $u_{2_e}$ values.

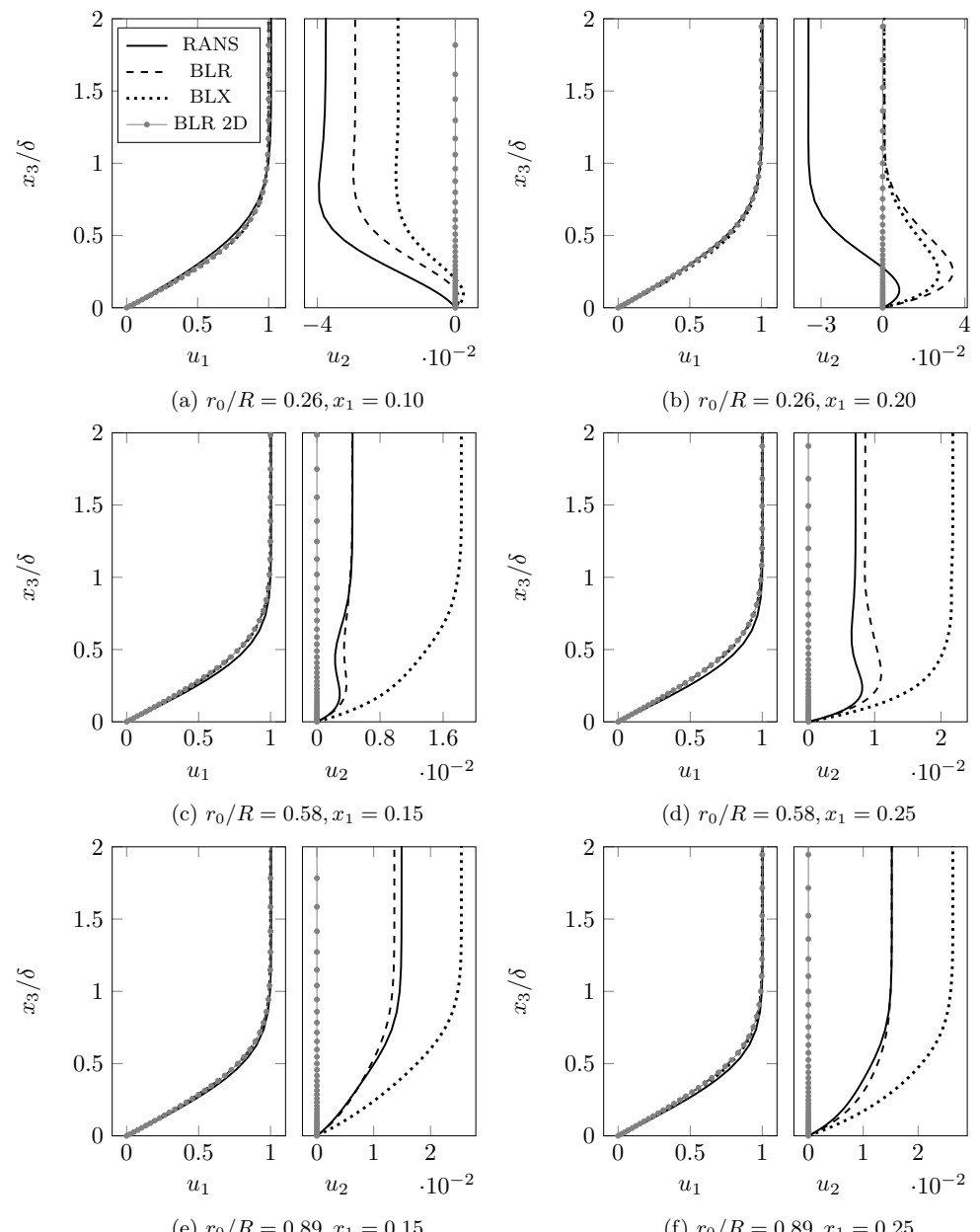

**Figure 6.** Boundary layer profiles for Geometry 1.

The BLR and BLX results for the spanwise velocity at the inner and middle radial parts of Geometry 2 (Figs. 7a, 7b, 7c, and 7d) in general do not follow the trend of the RANS results. An exception is the BLR spanwise velocity profile at $r_0/R = 0.58$ and $x_1 = 0.25$. As shown in Figs. 7a and 7b, the RANS profile presents an inversion of direction between 10 % and 20 % of the chord. This also occurs in a smaller extent at the inner radial position of Geometry 1 (Figs. 6a and 6b) where, at the near-wall

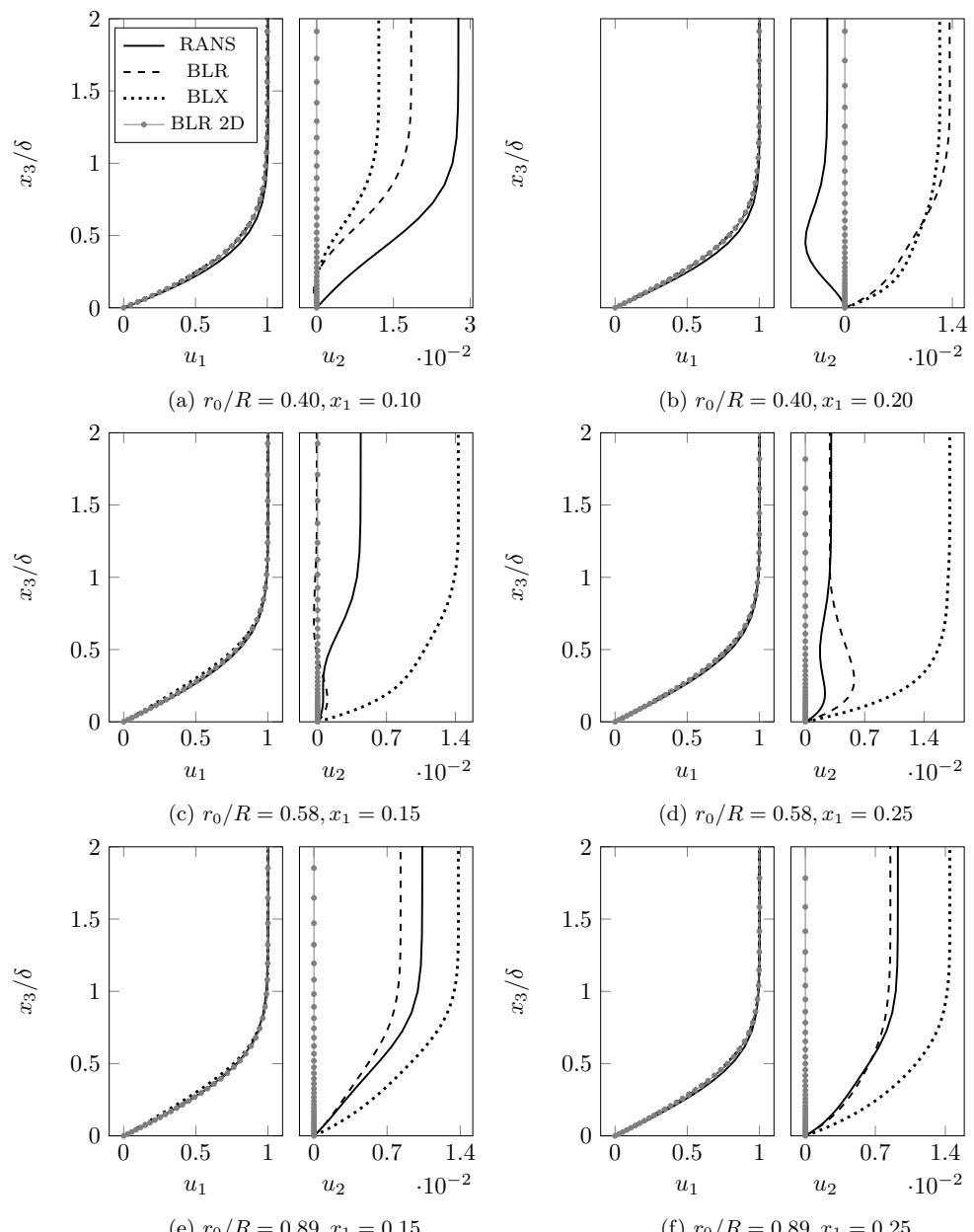

**Figure 7.** Boundary layer profiles for Geometry 2.

region, the spanwise velocity profile presents an inversion of direction. The fact that the inversion of the spanwise velocity
profile only occurs at the inner radial position of Geometries 1 and 2 may confirm the three-dimensional character of the flow
at smaller radii.

The effects of rotation on the spanwise velocity are investigated using the approach of Du and Selig (2000), in which the rotation speed is varied while the angle of attack is kept constant. This allows for segregating the effects of the variation of the spanwise velocity as well as Coriolis and centrifugal forces from those caused by the variation of the angle of attack. The selected rotation speeds are 5 %, 50 %, 100 %, and 150 % of that used in RANS ($0.64$ and $0.9$ rad $\cdot$ s$^{-1}$ for Geometries 1 and 2, respectively). The 5 % and 50 % cases account for the accelerating phase of the wind turbine, whereas the 150 % case is not in the normal operating range of most turbines but offers insight into how overspeed may impact transition.

Analysis of the data for Geometry 1 shows that the inviscid flow is accelerated in the $-x_2$ direction near the stagnation point due to a negative spanwise pressure gradient and the Coriolis force to a lesser extent. The dominant term of the latter is $-2\rho u_1 \Omega_3$ in Eq. (11), pointing in the $-x_2$ direction. After roughly 10 % of the chord, where the flow reaches its maximum streamwise velocity, the spanwise pressure gradient decreases substantially. Hence, the centrifugal force, with leading term $\rho \Omega_3^2 x_2$ in Eq. (11), and the inertial term with $\rho u_1^2$ in Eq. (12) overcome the Coriolis force and accelerate the flow in the $+x_2$ direction. For small radii, the Coriolis force tends to increase faster with the rotation speed than the centrifugal and inertial ones, impelling the flow in the $-x_2$ direction. The centrifugal and inertial forces tend to grow faster with $\Omega$ at the middle and outer parts of the blade, forcing the flow in the $+x_2$ direction.

Figure 8 presents the BLX spanwise velocity profiles for Geometry 1. Compared to an almost translatoric situation ($0.032$ rad$\cdot$ s$^{-1}$), rotation tends to accelerate the flow in the $x_2$ direction, driven by the centrifugal and inertial forces. Considering $r_0/R = 0.58$ and $0.89$, the spanwise velocity increases with $\Omega$ since the centrifugal and inertial forces grow faster at larger radii. At the inner radial position, the spanwise velocity decreases when $\Omega$ increases from $0.32$ to $0.96$ rad $\cdot$ s$^{-1}$ because the Coriolis force grows faster than its counterparts. These velocity profiles present inflection points, indicating the potential of crossflow instability. Inflectional profiles can also be observed at the inner radial position of Geometry 2.

The boundary-layer profiles for Geometry 2 are presented in Fig. 9. The airfoils of Geometry 2 sustain negative chordwise and spanwise pressure gradients over a larger chordwise extent compared to Geometry 1. Therefore, it is not possible to decouple a region where the pressure gradient is dominant from another in which rotation effects are preponderant. This fact makes the effects of rotation less clear than in the previous geometry. However, one can still observe the trend described in the theoretical analysis. At the downstream chordwise stations, the flow accelerates with $\Omega$ in the $-x_2$ at the inner locations and in $+x_2$ directions at the outer sections. An increase in the rotation speed tends to accelerate the flow in the $-x_2$ direction at $r_0/R = 0.58$. This fact indicates that the pressure gradient and Coriolis forces are more important than the inertial and centrifugal ones at this location. This trend remains for the downstream chordwise station since the negative spanwise pressure gradient does not vanish.

## 4.5 Transition prediction

The quasi-three-dimensional PSE model is applied to analyze the disturbance growth inside the boundary layer. The onset of transition is assumed to occur when the amplification factor $N$ based on the integral disturbance energy (Hanifi et al., 1994) reaches $N_{crit}$. This state corresponds to the appearance of the first turbulent spots. Although not representative of all atmospheric conditions, it is assumed $N_{crit} = 9$ in the current work to have a larger region of laminar flow in the RANS results,

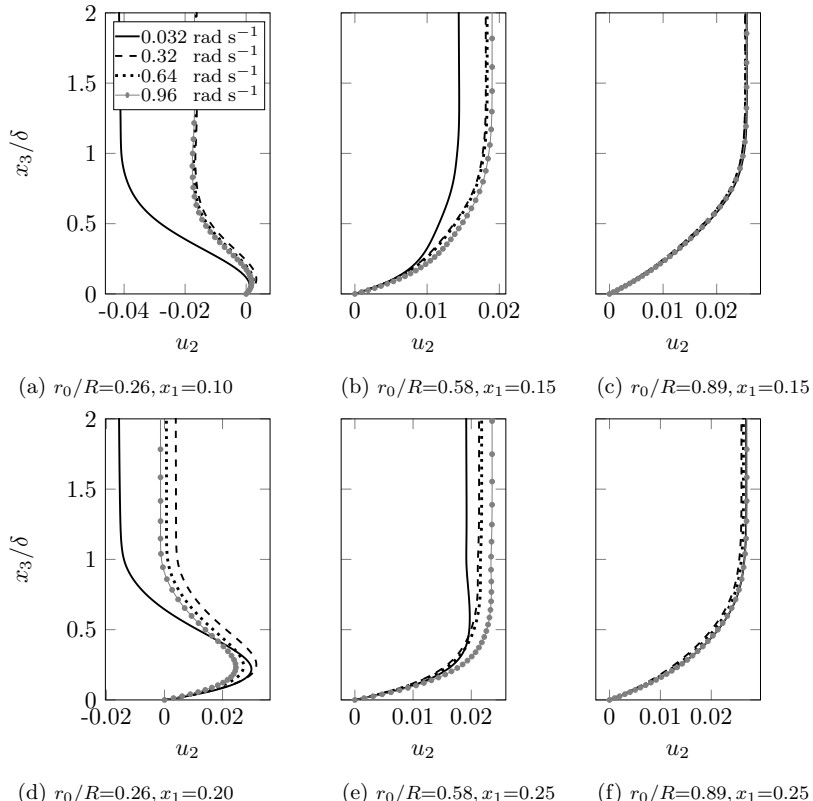

**Figure 8.** Spanwise velocity profiles for Geometry 1 for several rotation speeds.

allowing a more detailed comparison between the developed model and RANS. In the EllipSys3D code, when the $e^N$ method of Drela and Giles (1987) indicates that $N_{crit}$ was reached, the onset of transition is detected and the intermittency factor $\gamma$ starts to grow from zero in the laminar region to one in the fully turbulent flow (Özçakmak et al., 2020). As the transition location is not directly stored in RANS data, we choose to select a small value for this parameter ($\gamma = 0.01$ is selected) to
385 indicate the transition location.

The transition locations for Geometry 1 as a function of the radial position are presented in Fig. 10a. Transition is delayed as the radial position increases, which agrees with previous works that observed stabilizing effects of rotation for increasing radii (Du and Selig, 2000). PSER and RANS transition locations agree from $r_0/R = 0.68$ to the tip of the blade. For $r_0/R < 0.68$, PSER results indicate an earlier transition compared to RANS. This is due to the effects of the spanwise velocity and rotation, which are not considered in the EllipSys3D transition model. As shown in Section 4.4, the spanwise velocity reaches higher
values at lower radii. Moreover, the presence of a laminar separation bubble at the inner part of the blade increases the rotation effects because the Coriolis force passes to act in the same direction of the centrifugal one. Therefore, differences between transition locations from RANS and the developed model were expected to be larger at lower radii. Another conclusion is that considering three-dimensional and rotation effects leads to the prediction of earlier transition locations. The PSER 2D

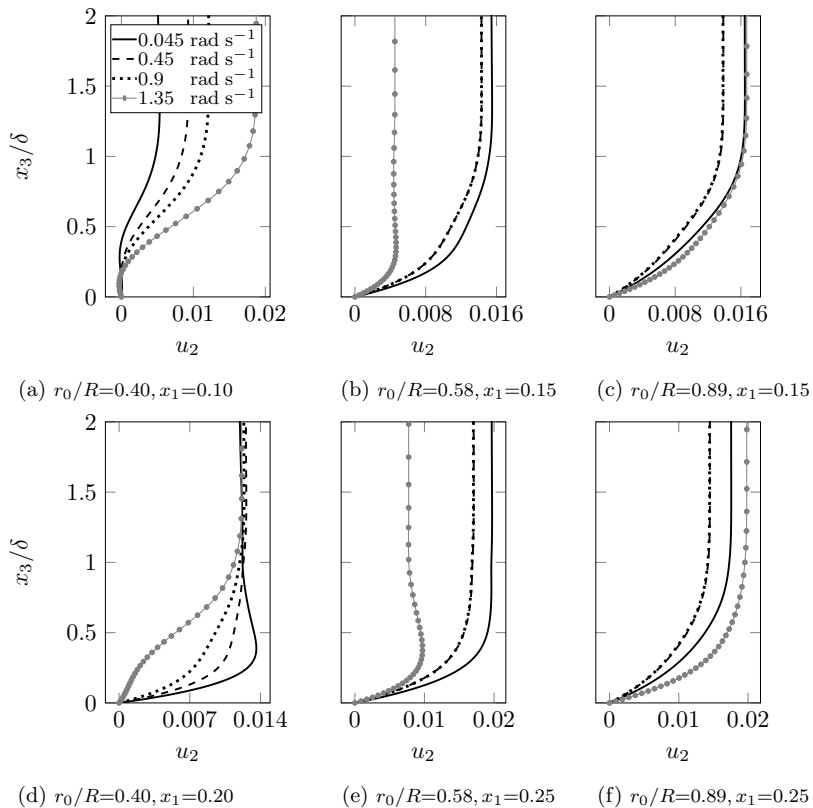

(a) $r_0/R{=}0.40, x_1{=}0.10$     (b) $r_0/R{=}0.58, x_1{=}0.15$     (c) $r_0/R{=}0.89, x_1{=}0.15$

(d) $r_0/R{=}0.40, x_1{=}0.20$     (e) $r_0/R{=}0.58, x_1{=}0.25$     (f) $r_0/R{=}0.89, x_1{=}0.25$

**Figure 9.** Spanwise velocity profiles for Geometry 2 for several rotation speeds.

transition locations, which do not consider 3D and rotational effects, are in close agreement with the RANS results, except at $r_0/R = 0.26$, where the former indicates transition slightly downstream. Concerning the PSEX results, earlier transition locations are obtained for $r_0/R \geq 0.58$ compared to RANS and PSER. This is likely due to the higher spanwise velocity found at these locations with the PSEX method. PSEX and PSER transition locations are close to each other for lower radial positions, probably because the differences between their predicted spanwise velocity profiles are smaller.

Figure 10b presents the transition locations for Geometry 2. PSER and PSER 2D results are in close agreement. This indicates that three-dimensional effects and rotation are likely not very important for this blade. As discussed Section 4.4, the pressure gradient seems to be more important than rotation effects in Geometry 2. PSER and PSER 2D present slightly downstream transition locations when compared to RANS. The PSEX transition locations are downstream of the PSER ones, possibly due to the weaker adverse pressure gradient in the $C_p$ distributions from XFOIL. The transition delay due to increasing radius is less significant in Geometry 2, probably because of the lower influence of rotation effects.

The PSER contours of $N$-factor as a function of the chordwise position and propagation angle $\Psi$ are shown in Fig. 11. $\Psi$ is the angle between the inviscid streamline and the perturbation wavevector (see Fig. 1). The dashed red line indicates the transition location. Considering Geometry 1 in Figs. 11a, 11b, and 11c, the region of critical $N$-factor is displaced in

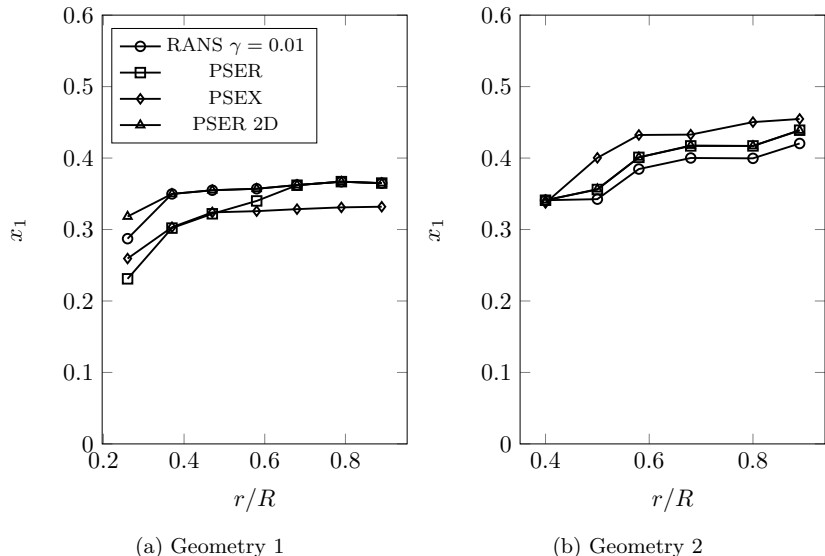

(a) Geometry 1      (b) Geometry 2

**Figure 10.** Transition locations.

the $-\Psi$ direction and it is less symmetrical at the inner radial location. The critical modes have $\Psi = -58°$, $-24°$, and $-6°$
at $r_0/R = 0.26$, 0.58, and 0.89, respectively. The lower $\Psi = -58°$ at $r_0/R = 0.26$ is possibly related to the stronger and
inflectional spanwise velocity occurring at this location, which makes transition more susceptible to oblique and crossflow
modes. Transition occurs significantly earlier at this position ($x_1 = 0.23$, compared to $x_1 = 0.34$ and 0.37 at $r_0/R = 0.58$
and 0.89, respectively). The PSER 2D contours of $N$-factor, shown in Figs. 12a, 12b, and 12c, are more symmetrical around
$\Psi = 0°$, with the critical modes having lower $|\Psi|$ ($\Psi = 17°$, 5°, and 4° for $r_0/R = 0.26$, 0.58, and 0.89, respectively). This
shows that the oblique critical modes obtained in the PSER results are caused by three-dimensionality and rotation.

Figures 11d, 11e, and 11f show that the PSER critical regions are more elongated in the $\Psi$ direction for Geometry 2,
indicating transition susceptibility to a broader range of waves. The critical modes have $\Psi = -12°$, $-16°$, and $-12°$ for
$r_0/R = 0.40$, 0.58, and 0.89. These waves are less oblique than those for Geometry 1, particularly at the inner radial location.
Notice that the BL profiles of spanwise velocity at this location (Fig. 7b) do not present an inflection point, making transition
via lower $|\Psi|$ modes more likely. Regarding the PSER 2D results, in Figs. 12d, 12e, and 12f, the regions of critical $N$-factors
are more centered around $\Psi = 0°$, with the critical modes for $r_0/R = 0.40$, 0.58, and 0.89 presenting $\Psi = 0°$. This means that
disregarding 3D and rotation effects in the mean-flow leads to 2D critical modes for Geometry 2.

Figures 13a, 13b, and 13c present the profiles of the perturbation of $u_1$ velocity of the modes leading to transition in
Geometry 1. The PSER and PSEX modes are in close agreement for the three radial positions, indicating that they predict the
same transition mechanism. At $r_0/R = 0.26$, these modes have a single peak, located at $x_3/\delta = 0.2$, associated with their high
$|\Psi|$ and the inflectional spanwise velocity (Fig. 6b). This indicates that transition may be triggered by oblique TS or crossflow
modes. The PSER 2D critical mode differs from the previous ones by presenting a near-wall peak, at $x_3/\delta = 0.1$, and having

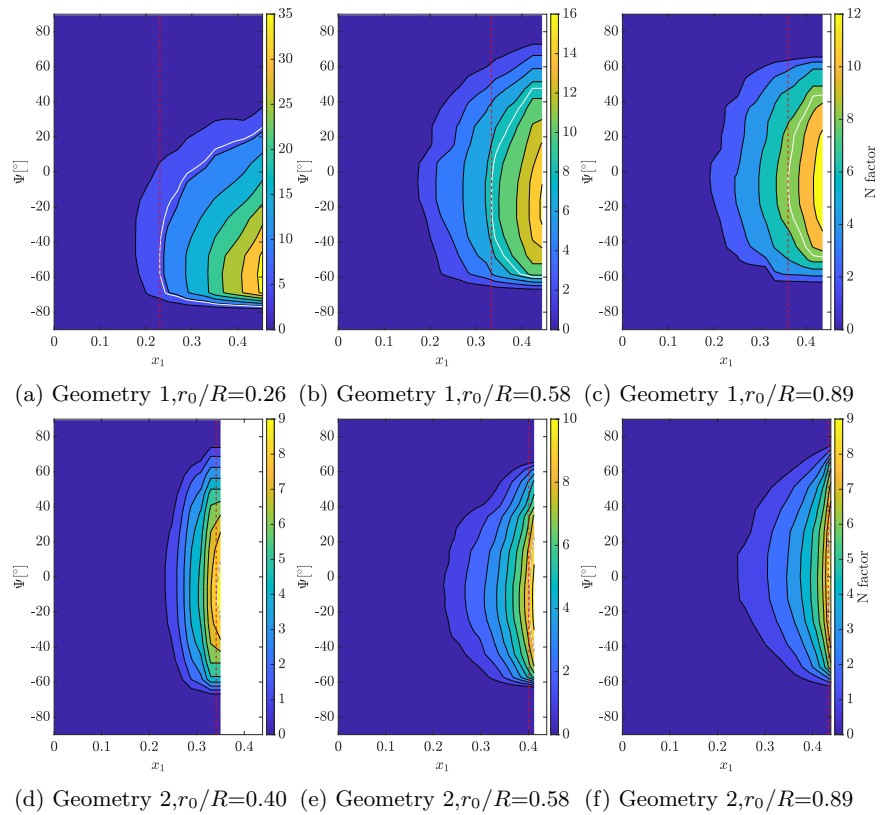

**Figure 11.** $N$-factor contours from PSER for three radial positions. The white line indicates the critical region, and the red dashed line shows the transition location.

a second lobe for $x_3/\delta > 0.7$ . At $r_0/R = 0.58$, the PSER and PSEX modes approach the PSER 2D one by developing a near-wall peak, although less important than the one at $x_3/\delta = 0.2$, and a second lobe for $x_3/\delta > 0.7$. The PSER and PSEX

modes finally converge to a 2D mode at $r_0/R = 0.89$, where they are in close agreement with the PSER 2D one. The latter is similar to a 2D TS wave, as also observed for $r_0/R = 0.58$. The appearance of near-wall peaks in the PSER and PSEX modes at $r_0/R = 0.58$ and 0.89 as well as the close agreement between these modes and the PSER 2D ones at $r_0/R = 0.89$ can be related to the amplification of 2D TS waves due to an adverse pressure gradient.

The results for Geometry 2 are presented in Figs. 13d, 13e, and 13f. As occurs for Geometry 1, the PSER and PSEX modes

agree for the three radial positions. They indicate double-peak modes, with maxima at $x_3/\delta = 0.1$ and 0.2. The former has a larger or similar magnitude compared to the latter. These modes are close to the PSER 2D ones except around $x_3/\delta = 0.2$, where the PSER and PSEX modes have more pronounced peaks. The presence of a peak at $x_3/\delta = 0.1$ for all radial locations is related to a strong adverse pressure gradient in Geometry 2. The second peak, at $x_3/\delta = 0.2$, seems to be associated with the obliqueness of the mode, having a larger amplitude for larger values of $|\Psi|$. A 2D TS mechanism seems to be more important in

Geometry 2 because the critical modes are closer to the PSER 2D ones, and the adverse pressure gradient is stronger. However,

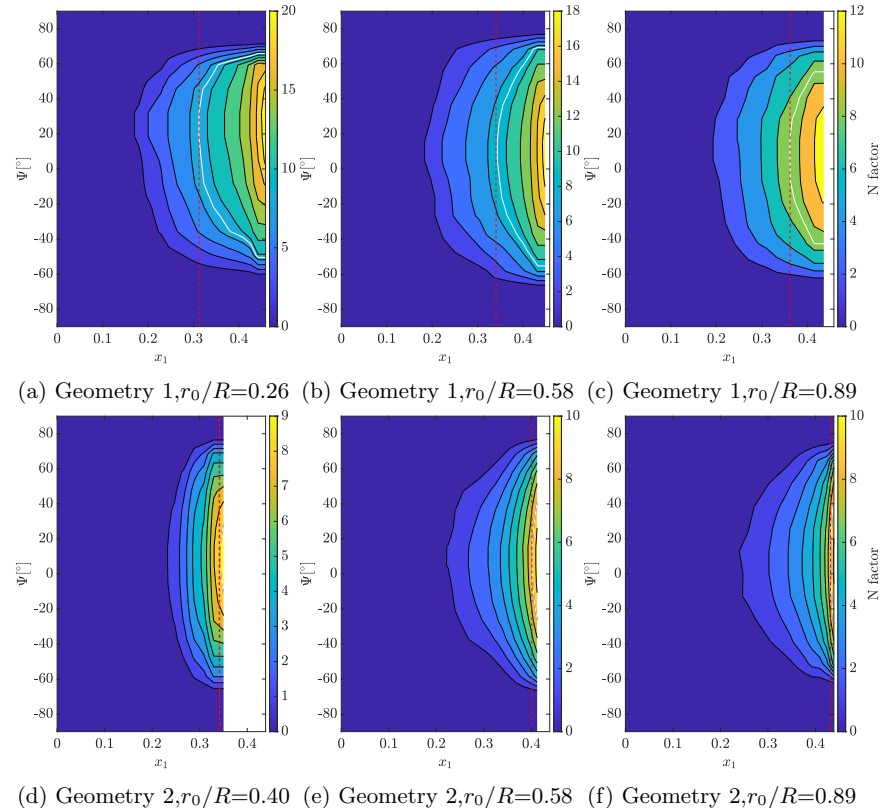

(a) Geometry 1,$r_0/R$=0.26  (b) Geometry 1,$r_0/R$=0.58  (c) Geometry 1,$r_0/R$=0.89

(d) Geometry 2,$r_0/R$=0.40  (e) Geometry 2,$r_0/R$=0.58  (f) Geometry 2,$r_0/R$=0.89

**Figure 12.** $N$-factor contours from PSER 2D for three radial positions. The white line indicates the critical region, and the red dashed line shows the transition location.

a mechanism related to oblique TS waves, engendered by 3D and rotation effects, appears to be more important for transition in Geometry 1. This is due to its larger sweep angle and region of favorable pressure gradient. Although the crossflow velocity profiles are inflectional, the magnitude of this velocity component is very low, of the order of 0.1 % of the freestream velocity, except for the inner radial location of Geometry 1, where it reaches 3.5 %. Thus excluding Geometry 1 at $r_0/R = 0.26$, a
crossflow transition mechanism is unlikely. Nevertheless, the effect of the spanwise velocity on transition cannot be neglected as it allows transition through oblique modes.

In the next, we analyze the effects of rotation on the transition locations. Figure 14a presents the PSEX transition locations as a function of the radial position and rotation speed for Geometry 1. The displayed trend indicates that an increase in the rotation speed shifts the transition location closer to the nose. In particular, the rise in $\Omega$ from 0.32 to 0.96 rad $\cdot$ s$^{-1}$ leads to transition
37 % earlier. The case corresponding to 5 % of the RANS rotation speed (not shown) did not present any mode reaching $N_{crit}$ further indicating the destabilizing effect of rotation. These effects occur through the Coriolis and centrifugal forces acting on the disturbances as well as through the variation of the spanwise velocity. The former seems to be preponderant since there is no significant variation in the spanwise velocity with $\Omega$ at $r_0/R = 0.89$, but transition occurs earlier regardless. There is a

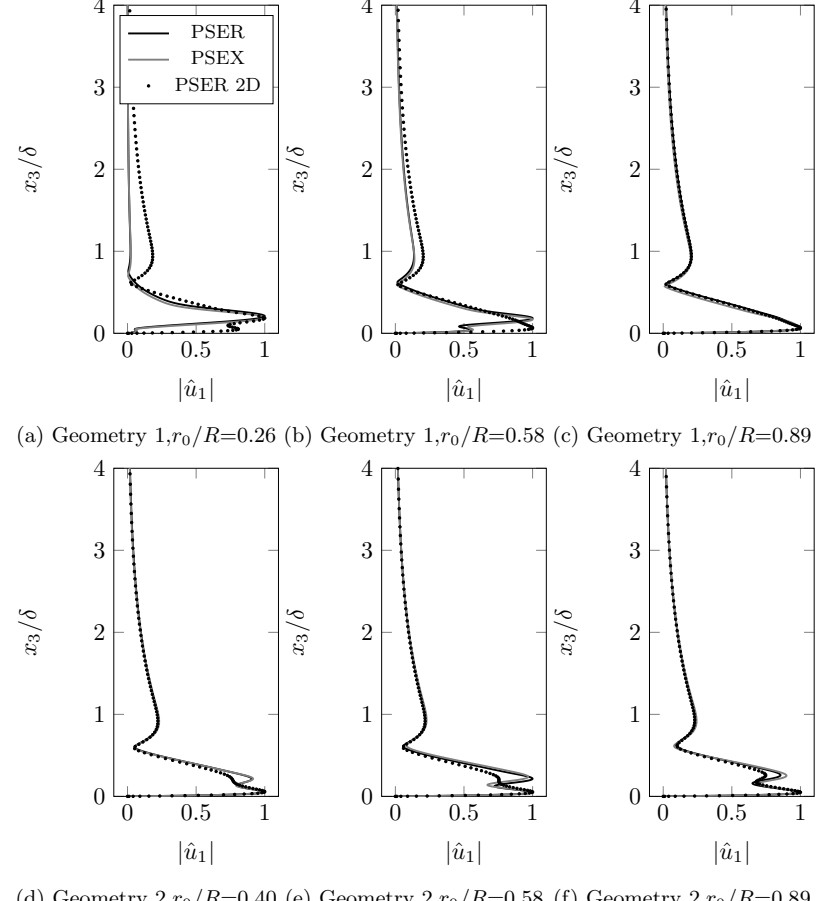

(a) Geometry 1,$r_0/R$=0.26 (b) Geometry 1,$r_0/R$=0.58 (c) Geometry 1,$r_0/R$=0.89

(d) Geometry 2,$r_0/R$=0.40 (e) Geometry 2,$r_0/R$=0.58 (f) Geometry 2,$r_0/R$=0.89

**Figure 13.** PSE results for the mode leading to transition.

delay in transition for increasing radius up to $r_0/R = 0.47$, where the Coriolis force is prevalent. Further increases in radius do
not significantly change the transition locations, indicating a balance between the rotation effects. The presence of a laminar
separation bubble for radial positions closer to the root can make the Coriolis force act in the same direction as the centrifugal
one. For higher radial positions and in the absence of separation, these two forces tend to balance each other.

Figure 14b portrays the results for Geometry 2. The increase in $\Omega$ plays a destabilizing role. This observation is supported
by the fact that the case with 5 % of the RANS rotation speed (not shown) presented no mode reaching $N_{crit}$. However, the
variation of $\Omega$ does not play a role as important as for Geometry 1. For instance, transition occurs 8 % earlier on average for an
increase in $\Omega$ from 0.45 to 1.35 rad·s$^{-1}$. The transition location moves less with the rotation speed for Geometry 2 because this
blade maintains a non-negligible pressure gradient over a larger chordwise extent, overtaking rotation effects. The fact that the
spanwise velocity in Geometry 2 varies more with $\Omega$ than in Geometry 1 with a smaller effect on transition corroborates this

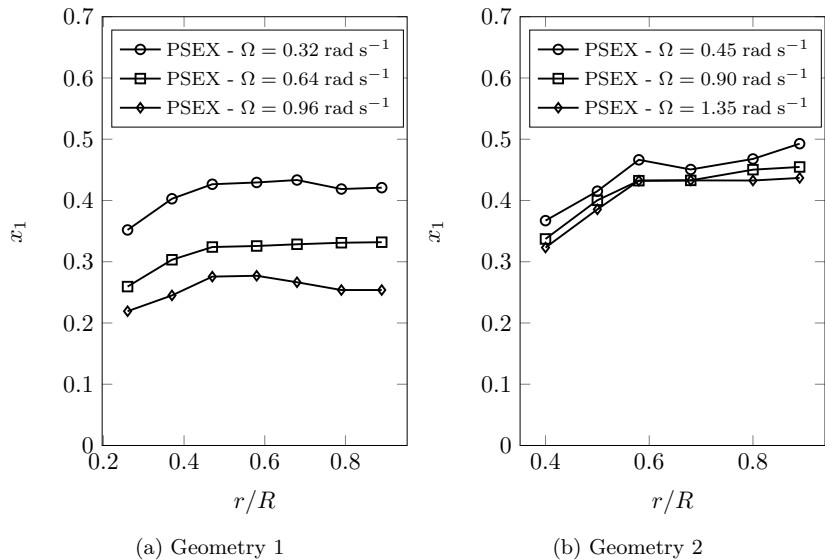

(a) Geometry 1         (b) Geometry 2

**Figure 14.** Transition locations for several rotation speeds.

claim. Transition is delayed when increasing the radius up to $r_0/R = 0.58$, a range along which the Coriolis force is dominant.
Only slight variations in transition locations occur after this radial position, pointing to a balance in the rotation effects.

The PSEX contours of $N$-factor at $r_0/R = 0.58$ for Geometries 1 and 2 are shown in Fig. 15. In the case of Geometry 1, as shown in Figs. 15a, 15b, and 15c, the increase in $\Omega$ forces the critical region towards lower $x_1$. This region lies mostly in the $-\Psi$ half-plane, meaning that the critical waves propagate towards the root of the blade. These modes present $\Psi = -25°$, $-24°$, and $-25°$ for $\Omega = 0.32, 0.64$, and $0.96$ rad·s$^{-1}$. For Geometry 2, in Figs. 15d, 15e, and 15f, we also observe the displacement of the
470 critical region to lower $x_1$ with the increase in $\Omega$. Moreover, the flat critical region extending from $\Psi = -60°$ to $40°$ obtained with $\Omega = 1.35$ rad·s$^{-1}$ shows that the higher rotation velocity allows transition through a broader range of disturbances. The critical regions are mostly located in the $-\Psi$ half-plane, indicating stronger transition susceptibility to waves traveling to the inner blade part. The critical modes present $\Psi = -16°$, $-15°$, and $-13°$ for $\Omega = 0.45, 0.9$, and $1.35$ rad·s$^{-1}$. The analysis of the full geometry indicates that the increase in $\Omega$ reduces the critical $|\Psi|$ in the region $0 \leq r_0/R \leq r$, where $r = 0.58$ and $0.5$
for Geometries 1 and 2. For larger $r$, the opposite occurs, i.e., rising $\Omega$ leads to increasingly oblique critical modes.

Figures 16a, 16b, and 16c show the PSEX profiles of the critical modes for Geometry 1. All modes collapse at the inner radial location, indicating that $\Omega$ does not alter the transition mechanism. The inflectional spanwise velocity profiles at this location (Figs. 8a and 8d) seem to render the transition mechanism, through oblique modes, quite robust to changes in $\Omega$. At $r_0/R = 0.58$ and $0.89$, the modes for $\Omega = 0.64$ and $0.96$ rad·s$^{-1}$ are in close agreement. However, the mode for $\Omega = 0.32$ rad·s$^{-1}$
differs from the previous ones by the presence of a near-wall peak. As already discussed, the mode shapes are closely related to their propagation angles, with higher-$|\Psi|$ modes occurring at locations of inflectional spanwise velocity and tending to have a single-peak like those at $r_0/R = 0.26$. Figures 16d, 16e, and 16f shows the results for Geometry 2. At $r_0/R = 0.40$, the increase in $\Omega$ reduces $|\Psi|$ and makes double-peak modes such as those for $\Omega = 0.45$ and $0.9$ rad·s$^{-1}$ become a 2D, single-peak

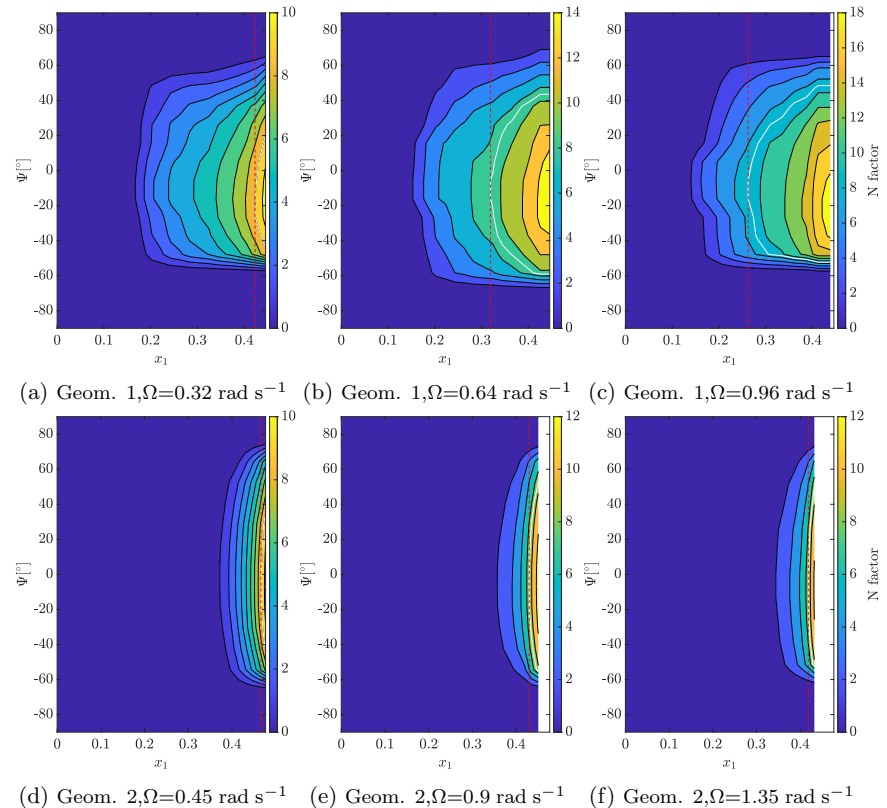

(a) Geom. 1,$\Omega$=0.32 rad s$^{-1}$  (b) Geom. 1,$\Omega$=0.64 rad s$^{-1}$  (c) Geom. 1,$\Omega$=0.96 rad s$^{-1}$

(d) Geom. 2,$\Omega$=0.45 rad s$^{-1}$  (e) Geom. 2,$\Omega$=0.9 rad s$^{-1}$  (f) Geom. 2,$\Omega$=1.35 rad s$^{-1}$

**Figure 15.** $N$-factor contours from PSEX at $r_0/R = 0.58$ for several rotation speeds. The white line indicates the critical region, and the red dashed line shows the transition location.

mode like the one for $\Omega = 1.35\,\mathrm{rad \cdot s^{-1}}$. At $r_0/R = 0.58$, all modes collapse and present double peaks. At the outer radial location, the mode for $\Omega = 0.45\,\mathrm{rad \cdot s^{-1}}$ is nearly 2D, and the rise in $\Omega$ increases its obliqueness (i.e., increases $|\Psi|$). The modes for higher $\Omega$ are in close agreement at this location. In Geometry 2, the adverse pressure gradient is more important, and transition is more susceptible to modes closer to 2D TS waves with near-wall peaks. The increase in the rotation tends to prompt these 2D modes at low radial locations, while it makes the critical modes more oblique at higher radii.

## 5  Conclusions

A framework for transition prediction applicable to flows over wind-turbine blades is developed. The method, which comprises a boundary-layer model and the PSE, accounts for effects of the quasi-three-dimensional flow and the blade rotation. It aims to provide more reliable transition predictions without requiring three-dimensional simulations. Using the developed method, we have analyzed the role of flow three-dimensionality and rotation on the transition onset over two geometries.

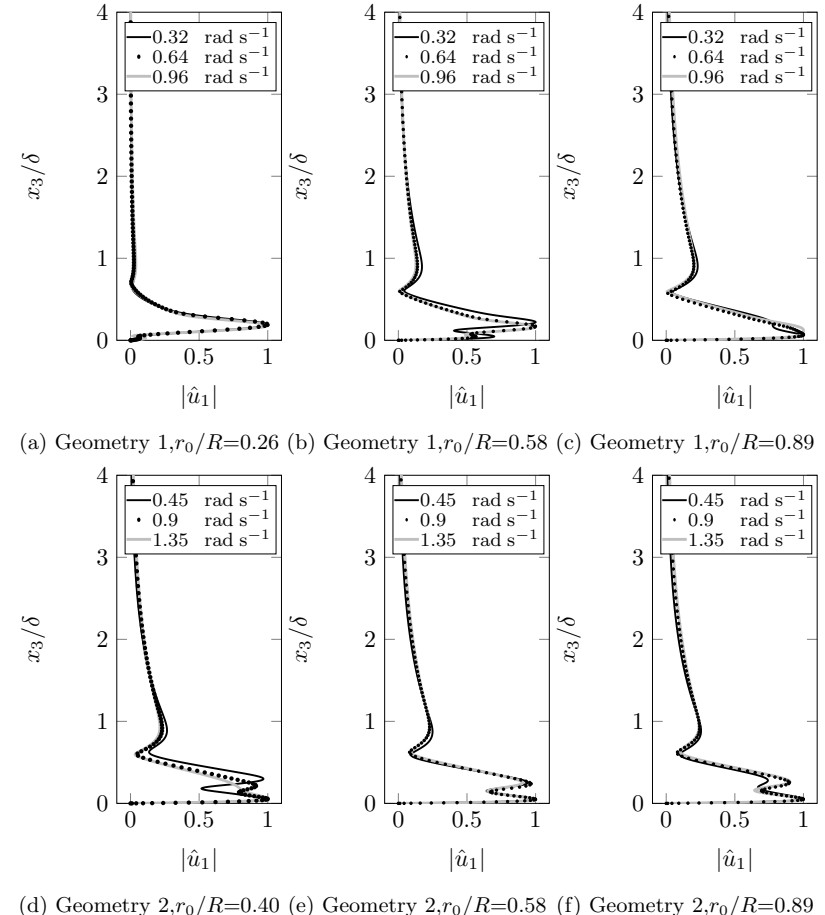

(a) Geometry 1,$r_0/R$=0.26 (b) Geometry 1,$r_0/R$=0.58 (c) Geometry 1,$r_0/R$=0.89

(d) Geometry 2,$r_0/R$=0.40 (e) Geometry 2,$r_0/R$=0.58 (f) Geometry 2,$r_0/R$=0.89

**Figure 16.** PSEX results for the mode leading to transition for several rotation speeds.

The developed method provides accurate chordwise velocity profiles and, for locations not too close to the root of the blade
and stagnation point, spanwise velocity. The flow is highly three-dimensional close to the root of the blade, reducing the
accuracy of a quasi-three-dimensional approach. The spanwise velocity obtained with the model better agrees with RANS for
geometries respecting the conical-wing approximation. Some of the spanwise velocity profiles contain inflection points, which
may allow crossflow instability, not considered in two-dimensional transition models. Rotation was shown to accelerate the
flow towards the tip of the blade in the developed flow region, while the opposite occurs near the stagnation point.
Transition locations from the $e^N$ method implemented in the EllipSys3D RANS code closely agree with those from the PSE
analysis of a 2D mean-flow without rotation. RANS transition locations are close to those from the model developed in this
work in places where 3D and rotation effects are low. This occurs for Geometry 2 and higher radial positions in Geometry 1.
However, results of the RANS transition model and the 2D approach deviate from those from the new approach for locations
from the root to approximately 58 % of the radius of Geometry 1, where 3D and rotation effects are important. At these

505 locations, the combined influence of three-dimensionality and rotation leads to earlier transition onsets. These effects make transition occur through oblique modes, which have single peaks and are not predicted with the 2D approach. The oblique modes appear in locations where the spanwise velocity profile is inflectional, raising the possibility of being related to crossflow instability. However, except for the inner radial location of Geometry 1, the magnitude of the crossflow velocity seems to be too low to trigger crossflow transition. The single-peak modes may be very oblique TS waves. For larger radial positions, the flow

tends to be more two-dimensional, and the adverse pressure gradient is more important. Thus the critical modes become less oblique and develop features of 2D TS waves, such as a second peak near the wall. Finally, it is also shown that the increase in the rotation speed, through the modification of the spanwise velocity and the increase in the Coriolis and centrifugal forces, seems to shift the transition location closer to the leading edge.

In order to better understand the transition process over the rotating blades and validate the prediction of the presented

approach, in-depth investigation through DNS simulations and detailed experimental works are desired.

*Code and data availability.*  Part of the codes and data employed/developed is available upon direct request with the corresponding author.

*Author contributions.*  TF implemented the models, performed the analysis, and wrote the final version of the manuscript. ML developed the model, obtained part of the results, and wrote the first version of the document. NS and FZ performed the RANS simulations using the EllipSys3D code. AH and DH developed the NOLOT PSE code (among other researchers), supported the analysis, provided useful

discussions, and contributed with critical feedback. All authors reviewed the manuscript.

*Competing interests.*  The authors declare that they have no conflict of interest.

*Acknowledgements.*  This research has been supported and funded by the Strategic Research Area initiative StandUp for Energy.

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
