# Peer review of "A simplified model for transition prediction applicable to wind-turbine rotors"

_Wind Energy Science, 2020_

## Referee Comment (RC1) · Anonymous Referee #1 · 14 Dec 2020

Paper: wes-2020-107

Title: Low-order modelling for transition prediction applicable to wind-turbine Rotors

Authors:  Th. Fava et al.

**General:**

The authors present results form a simplified model for use of transition prediction for wind turbine rotors. This topic is of considerable current interest not only to develop more adapted aerodynamic profiles to increase aerodynamic efficiency but also from a more scientific point of view to detect to main mechanism for transition from the laminar to turbulent state.

However, it is no always easy to follow the text. Authors should consider:

a) A list/table of abbreviations,
b) Improving the style of writing by careful discussion with a native speaker,
c) Being more exact in the wording,
d) Shortening the text.
e) Include references from recent experiments:
(10.5194/wes-5-1487-2020,  10.3390/en12112102 )

**Specific:**

Title: what is meant by "Low order"? Order in what?

What about "A simplified …"

Line 1: "onset ..." Do you mean the "critical" point, where damping becomes negative first? Or do you mean "start of fully turbulent" region by choice of N?

Line 14. "reasonable accuracy" is not a scientific term. Use: accuracy in numbers instread, pp% for  example

Line 19 ff (Intro) so e) from above

Line 21: typo

Line 66 to 69: "However, … is expected to be more accurate ..." Why?

Line 85: Usually, when using body-fitted coordinates, a metric TENSOR appears ($g_{ij}$). Please show its relation the metric VECTOR you are using.

Line 96: "costly" Are we talking about € or $? Please be more accurate in wording and comparing typical amount of CPU hrs.

Line 228: please give of precise definition of intermittency ($\gamma$)

Table 2: Geometry 2: "Varying" is not sufficient. Please state at least names.

Line 268/269: "The discrepancies … non-respect … these locations." This sentence is hard to understand. Please improve.

Line 337: N = 9. Why did you choose this very specific value more appropriate for WIND TUNNEL experiments? As you may know, wind turbines operate in very different inflow conditions. Please improve.

Line 340: I do not understand why "γ=0,01" should correspond to N=9. Please explain.

Line 355 ff and Fig. 11: I do not understand your explanation why PSEX/PSER group on one side and PSER 2D/RANS group on the other deviate so much. Instead of a description only, give more possible physical reasons.

Line 389 ff and Fig. 15:  I'm not sure if I have fully understood your explanation., If you are changing \omega (!) only by a factor of 3, tip-speed-ratio and angles of attack may vary as well that strong, so that your blade fully falls out of a meaningful operating range. On the other side transition location "only" varies by a factor less that two. Please explain in more detail.

Line 392: "accelerates transition". I think "accelerate" is not the right expression. What about "shifts the transition location closer to the nose"?

Line 457: "reliable estimate". Again, please state accuracy of your model more quantitavley

---

## Referee Comment (RC2) · Anonymous Referee #2 · 15 Dec 2020

This study was aimed at developing a tool for predicting boundary layer transition over wind turbine rotor blades. It is a pleasure to see that this approach, which has been seen a fair amount of success in the aerospace sector being extended for further applications. Despite, being of a lower-order this method is elegant and the flow physics is better represented, as opposed to the more computationally expensive, higher-order RANS methods which in fact have been demonstrated to be inferior here. The parametric analysis has helped in systematically assessing the effect of different conditions on the transition mechanisms that operates in such a complicated flow. Whereby, both two-dimensional and three-dimensional modes appear to operate. The analysis of the two geometries shows that there are still further potential benefits achievable through shape optimisation to delay transition and this kind of tool will be beneficial for such

exercises. Despite the few comments and remarks that follows below this manuscript is strongly recommended for publication.

Further comments:

1. Introduction – The linear stability of the flow over swept wings could have been further reviewed considering that the 3D boundary layer and PSE formulation was initially made for these kind of problems.

2. Equation 29 – subscript 1 missing for the streamwise coordinate and $\gamma$ has been used both for the frequency and as intermittency factor in the RANS modelling.

3. Overall, the discussion is quite thorough, however a bit prolix. For instance, a lot of effort has been placed in describing the shape of the velocity profiles but what are their implications in terms of boundary layer transition. As already reported on swept wings the highly inflectional nature of the profile of the transverse component (in this case $u2$), in the in-board region in figure 6(a) and 6(b) is also an indication of the potential crossflow instability. In fact, there is very little discussion on the behaviour of the mean flow and the possible route to transition, mainly focussed towards the end of section 4.2.

4. Similarly, the discussion on the flow over geometry 1 and geometry 2 are completely segregated. Since the topology of the flow is quite different from each other it will be interesting to compare them right from the beginning and this will already set the scene for how the modification of the mean flow by varying different parameter will favour a particular route to transition.

5. The sentence starting at line 349 and ending at line 350 is a bit of a contradictory statement.

6. Line 353 – "These differences arise from pressure distribution from XFOIL not exactly matching those from RANS, although they are close to each other". Any idea why there is this mismatch?

7. Figures 10 and 11 can be combined and similarly figures 15 and 16

8. Figures 12 and 17 can be rearranged so that they do not occupy a full page.

9. The angle $\eta$ could be described in the schematic in Figure 1 for readers who are not used to the 3D flow topology, may be just sketch of the flow topology such as the development of the skin friction lines. In fact, why not show the skin friction lines from the RANS simulation also to complement some of the arguments about the three-dimensionality being more pronounced on some part of the rotor.

10. The sentence starting from line 402 could be rephrased to be more explicit.

11. The splitting of the near-wall lobe of the TS eigenfunction is also observed in the presence of the large adverse streamwise pressure gradient; therefore it might be worth tying this with the strong 2D mode amplification.

12. The attachment line has been mentioned on quite a few occasions however there is no mention of whether it is below the threshold for contamination, keeping in mind that the leading edge radius of curvature can be quite considerable in the inboard region of the rotor.

12. Although the method developed here will be useful in the design and optimisation of wind turbine rotor blades, the linear PSE has its limitations which D. Henningson and A. Hanifi will definitely agree, therefore it might be worth mentioning those, just to keep the reader informed and avoid any bias within the transition community.

---

## Referee Comment (RC3) · Anonymous Referee #3 · 7 Jan 2021

**Review of the article entitled « Low-order modeling for transition prediction applicable to wind-turbine rotors » by T. Fava et *al.* for Wind Energy Science.**

**General opinion:**

This article presents a simplified method to predict/analyze the onset on laminar to turbulent transition on a fully 3D wind turbine blade. First the boundary layer equations are solved using an approximation for the external (ie at the boundary layer edge) spanwise velocity. Secondly, the stability of the obtained boundary layer mean velocity profiles is analyzed using a Parabolized Stability Equation (PSE) approach including rotation terms. This "tool" appears as very powerful since it only requires pressure distribution on dedicated spanwise sections which can easily be obtained with a code such as Xfoil based on panel method. Stability and transition prediction results are obtained for two blade geometries and compared to RANS computations integrating a database transition prediction tool. The influence of the rotating velocity on stability and transition location is also investigated. These results are very interesting and convincing.

Moreover, it should be noticed that the article is well written and organized so that it is really pleasant to read. For these reasons I strongly support the article for publication.

Below is a list of remarks/suggestions the authors should consider before publication.

**Specific remarks:**

L38-40: It is written that the PSE model are computational costly so not well suited for design. Nonetheless, the proposed model is based on PSE approach?

L50: It is mentioned that the reference RANS solver (EllipSys3D) integrate a transition prediction tool based on database method. Which one? Are there any references available? In the previous sentence, Sorensen 2009 is given as a reference but this paper deals with gamma-Retheta method which is not a database method. Additionally, since RANS results will be used as comparison for transition prediction, does the database integrates 3D effect and/or rotational effects?

L73: The equations (for the mean flow as well as for the fluctuations) are given in the general case of compressible flow including the equation for the temperature (or the energy). Is there any interest in considering the general compressible case or could it be restricted to simplified incompressible formulation? For an angular velocity of omega=1 rad/s and a radius of r=100m the azimuthal velocity at the tip will be around 100m/s ie a Mach number of 0.3 which is the usual limit to separate incompressible to compressible regimes. Additionally, the RANS code is incompressible (mentioned Section 4, p10).

L82: cp and gamma have already been used before. cp at the end of the abstract to refer to pressure coefficient and gamma in the gamma-Retheta model (ie referring to the intermittency function). It will be used again at the bottom of page 10 to refer to the intermittency factor. Additionally, page 9, equation 29 gamma is used as the angular frequency of the disturbances. A list of symbols at the beginning of the article will be helpful.

L123: It is mentioned that the pressure can be obtained from the velocity and temperature variables. For me one quantity is missing: the density or the total pressure.

L179-180: The PSE is derived from the continuity, s momentum (better suited), energy and state equations.

L198 (Eq 27): Even though periodicity is assumed in $x_2$ direction, 'q' and 'qtilde' depends on $x_2$. Same remark for eq 28.

L205: Already mentioned but the angular frequency is quoted as gamma which has already been used as the intermittency.

L217 (Eq33): the density fluctuation should also tends to 0 (or better be bounded) far away from the wall ($x_3 \rightarrow \infty$)

L221: The definition of the N factor should be given here. How is it related to the amplitude function of the disturbances? Moreover it is often specified that it is an envelope N factor but never said that the envelope is a local maximum on frequency (I guess).

L255: It is mentioned that cp distributions are approximation/differ from RANS ones. It would be interesting to illustrate (quantify) this discrepancy on a figure all the more than in the following this argument is reiterated to justify the differences between BLX and RANS velocity profiles L307 as well as on transition locations L353 and L357.

L284: "Although exhibiting higher values, the BLR and BLX profiles of spanwise velocity present the same shape of those from RANS." I do not understand this sentence since in Fig 6b, RANS provide a u2 profile lower than zero (except close to the wall) while the u2 profile provided by BLR and BLX is positive.

L294: "This is similar to what was observed in Geometry 1 and may indicate the three-dimensional character of the flow at lower radii". Not agree, related to the previous remark.

L337: The threshold value for eN method has been set up to 9 which is a typical value considering flight tests and quiet wind tunnel tests. Is this value appropriated for wind turbine blade application in the 'atmospheric boundary layer'?

L348: The PSE RANS results (not shown). It is unfortunate, since these results are the one to be considered as the reference to validate your approach. Is it possible to perform laminar RANS computations switching off the database transition prediction? Or rising the value of the transition threshold to obtain a laminar extend as long as possible in the RANS computation in order to analyze its stability with PSE? If not at least compare the evolution of the N factor between RANS and BLX for the first per cent of chord.

L384: "However, the modes tend to have a single-peaked structure at r0/R = 0.26, associated with their high propagation angle (in absolute value)." This has to be related to the spanwise mean velocity profile (fig 6a and b) which is inflectional (ie sensitive to CF like disturbances)

L434: "The single-peaked modes observed at r0/R = 0.26 for Geometry 1 and r0/R = 0.40 for Geometry 2 (! = 0.45 rad·s−1) might represent an intermediate stage between a TS and crossflow transition". Is there a shift/reduction of the frequency of the instability responsible for transition onset (the one reaching the Ncrit).

---

## Author Comment (AC1) · 21 Feb 2021

**Response to Reviewer's Comments concerning wes-2020-107**

Thales Fava    Mikaela Lokatt    Niels Sørensen

Frederik Zahle    Ardeshir Hanifi    Dan Henningson

We would like to thank the reviewer for careful and thorough reading of this paper. Our response follows.

**Referee #1 Comments**
* * *
**General:**

**Comment:** The authors present results form a simplified model for use of transition prediction for wind turbine rotors. This topic is of considerable current interest not only to develop more adapted aerodynamic profiles to increase aerodynamic efficiency but also from a more scientific point of view to detect to main mechanism for transition from the laminar to turbulent state. However, it is no always easy to follow the text. Authors should consider:

- a) A list/table of abbreviations,

- b) Improving the style of writing by careful discussion with a native speaker,

- c) Being more exact in the wording,

- d) Shortening the text.

- e) Include references from recent experiments: (10.5194/wes-5-1487-2020, 10.3390/en12112102 )

**Answer:**

- a) The authors agree that a table with abbreviations would make it easier for the reader to follow the text. We have included this feature as Table 3 at the beginning of the Results section (Section 4.1) in the revised document.

- b), c), and d) The style of writing has been improved. The manuscript has been revised to make the text exact and concise.

- e) The authors appreciate the suggestion from the referee. The mentioned references have been added to the list of literature and can be seen in the second paragraph of the Introduction section as well as in other parts of the revised document.
* * *
**Specific:**

- **Comment:** Title: what is meant by "Low order"? Order in what? What about "A simplified . . ."
  **Answer:** We understand that the use of the wording "Low-order modelling ... " in the title may confuse. We have changed it to "A simplified model ... ", as suggested by the referee since this is the intended meaning.

- **Comment:** Line 1: "onset ..." Do you mean the "critical" point, where damping becomes negative first? Or do you mean "start of fully turbulent" region by choice of N?
  **Answer:** "Onset of transition" is used meaning the position where the first turbulent spots

appear. This is supposed to happen at the location where the $N$ factor reaches a critical value (9 in the current case). This has been made clear in the first paragraph of Section 4.5 of the revised version: "The onset of transition is assumed to occur when the amplification factor $N$ based on the integral disturbance energy reaches $N_{crit}$. This state corresponds to the appearance of the first turbulent spots."

- **Comment:** Line 14. "reasonable accuracy" is not a scientific term. Use: accuracy in numbers instead, pp% for example
  **Answer:** The authors agree that a more appropriate expression should be used to denote the accuracy of the model. The model is accurate (compared to the RANS results) to predict the chordwise velocity profiles and, for regions not too close to the root of the blade and stagnation point, also the spanwise velocity profiles. Concerning the transition locations, it is not possible to state the accuracy, since the PSE analysis of the RANS base-flow, which would be the numerical reference for comparison of transition locations, does not yield reliable results (we believe that this is because the mean-flow derivatives computed in the post-processing step are not smooth enough). Thus we have opted to change the sentence in the Abstract of the revised document to: "The BL model allows an accurate prediction of the chordwise velocity profiles. Further, for regions not too close to the stagnation point and root of the blade, profiles of the spanwise velocity agree with those from Reynolds-averaged Navier-Stokes (RANS) simulations." We have also added to the Abstract the sentence: "The developed method, which accounts for these effects, predicted an earlier transition onsets in this region (e.g. 19 % earlier than RANS at 26 % of the radius for the constant-airfoil geometry) and shows that transition may occur via oblique modes."

- **Comment:** Line 21: typo
  **Answer:** "aerodynamiscists" has been corrected to "aerodynamicists"

- **Comment:** Line 66 to 69: "However, ... is expected to be more accurate ..." Why?
  **Answer:** The integral boundary-layer equations require closure relations which are found through empirical relations [4]. Therefore, we believe the differential form of the boundary-layer equations delivers a more accurate solution. We have changed the text in the first paragraph of Section 2.1.1 of the revised article to: "However, a differential formulation is expected to be more accurate than its integral counterpart because the latter requires closure relations which are found through empirical relations [4]. For this reason, a differential formulation is selected in the present case."

- **Comment:** Line 85: Usually, when using body-fitted coordinates, a metric TENSOR appears (gij). Please show its relation the metric VECTOR you are using.
  **Answer:** Here, we are using an orthogonal curvilinear coordinate system. Therefore, the metric tensor is represented as the Lamé coefficients $h_i$ (Eq. 5), where $h_i^2 = g_{ii}$. Note that since the coordinate system is orthogonal $g_{ij} = 0$ for $j \neq i$. This has been clarified in the second paragraph of Section 2.1.1 of the revised version: "Moreover, $\rho, p$, and $T$ denote density, pressure, and temperature, whereas $\mathbf{u}$ and $\mathbf{\Omega}$ represent velocity and rotation, respectively. $h_i$ are the Lamé coefficients, where $h_i^2 = g_{ii}$ and $g_{ij}$ is the metric tensor. Note that since the coordinate system is orthogonal $g_{ij} = 0$ for $j \neq i$."

- **Comment:** Line 96: "costly" Are we talking about € or $? Please be more accurate in wording and comparing typical amount of CPU hrs.
  **Answer:** Costly here is referring to the number of CPU hours. We have not performed a comparison between the computational cost of a 2D and a 3D BL code mainly because, from the authors' experience, the 2D version yields more accurate results. However, we estimate that the computational cost of solving the 3D BL equations in terms of CPU hours (for a serial code) and memory should scale with the number of grid points in the spanwise direction. This point has been clarified in the first paragraph of Section 2.1.2 of the revised article to: "A 3D discretization can result in a solution procedure that is costly in terms of computational capacity and CPU time."

- **Comment:** Line 228: please give of precise definition of intermittency $(\gamma)$
  **Answer:** $\gamma$ is defined as

$$\gamma = 1 - \exp\left\{-(x - x_{tr})^2 \left(\frac{U_{e,tr}}{\nu}\right)^2 \hat{n}\sigma\right\}, \text{ for } x \geq x_{tr}, \qquad (1)$$

where $x$ is the chordwise position (measured from the stagnation line, $x_{tr}$ is the chordwise position of the transition onset, $\nu$ is the kinematic viscosity, $\sigma$ is the spot propagation rate, $\hat{n}$ is the nondimensional spot formation rate, and $U_{e,tr}$ is the edge velocity at the chordwise position of the transition onset [6]. For laminar flow, i.e., $x < x_{tr}$, $\gamma = 0$, and for fully turbulent flow, $\gamma = 1$. This definition has been included in the first paragraph of Section 4 of the revised document.

- **Comment:** Table 2: Geometry 2: "Varying" is not sufficient. Please state at least names.
  **Answer:** Geometry 2 corresponds to the blade of the DTU 10 MW Reference Wind Turbine [1]. The FFA-W3-241 airfoil was used from 2/3 of the radius to the tip of the blade. From 2/3 of the radius to the root, the thickness of the mentioned airfoil was increased. This information has been included in Table 2 of Section 4.1 of the revised document.

- **Comment:** Line 268/269: "The discrepancies ... non-respect ... these locations." This sentence is hard to understand. Please improve.
  **Answer:** We have changed the sentence in the second paragraph of Section 4.3 to: "The differences between the EVM and RANS results are larger at the inner radial position and close to the stagnation point. The reason is that the approximation for the spanwise pressure gradient given by Eq. (12) is more accurate at large radii and chordwise positions. This approximation relies on the assumption of $C_p$ being constant over conical lines, which may not be respected at the mentioned locations due to the strong variation of the geometry in the radial direction and the flow three-dimensionality."

- **Comment:** Line 337: N = 9. Why did you choose this very specific value more appropriate for WIND TUNNEL experiments? As you may know, wind turbines operate in very different inflow conditions. Please improve.
  **Answer:** We agree with the referee that $N = 9$ represents transition in an environment with very low turbulence intensity (0.07 % according to Mack's relation [5]), not representative of all atmospheric conditions. This value was selected in order to have a larger region of laminar flow in the RANS results, allowing a more detailed comparison between transition results from the developed model and RANS. This information has been included in the first paragraph of Section 4.5 of the revised article: "Although not representative of all atmospheric conditions, it is assumed $N_{crit} = 9$ in the current work to have a larger region of laminar flow in the RANS results, allowing a more detailed comparison between the developed model and RANS."

- **Comment:** Line 340: I do not understand why "$\gamma = 0.01$" should correspond to N=9. Please explain.
  **Answer:** The transition method in the EllipSys3D RANS code is based on the $e^N$ model of Drela & Giles [2]. When the $N$ factor reaches a critical value (9 in the current case), it is assumed that the flow starts transitioning to turbulent. At that location, the turbulence production term in RANS is turned on. This term is multiplied by $\gamma$ whose value increases from zero to one along the chord following a given relation. The value of $\gamma$ is a function of the chord and span location and is a result of the RANS simulations. The transition point at each spanwise section corresponds to the point where $\gamma$ first deviates from zero. To extract the location of transition from the RANS data, we need to find those points. Due to the limited resolution in RANS simulations, we have selected location of $\gamma = 0.01$ to be the first point its value deviates from zero (the transition point). The following information has been included in the first paragraph of Section 4.5 of the revised article: " In the EllipSys3D code, when the $e^N$ method of [2] indicates that $N_{crit}$ was reached, the onset of transition is detected and the intermittency factor $\gamma$ starts to grow from zero in the laminar region to one in the fully turbulent flow [7]. As the transition location is not directly stored in RANS data, we choose to select a small value for this parameter ($\gamma = 0.01$ is

selected) to indicate the transition location."

- **Comment:** Line 355 ff and Fig. 11: I do not understand your explanation why PSEX/PSER group on one side and PSER 2D/RANS group on the other deviate so much. Instead of a description only, give more possible physical reasons.
  **Answer:** We have found an error in the code which prints the transition locations from the RANS and PSER 2D approaches. In the revised results, the agreement between these two methods is better, and they are closer to the PSEX and PSER results. The corrected results have been included in the revised manuscript and can be seen in Fig. (1).

[Figure]

(a) Geometry 1          (b) Geometry 2

Figure 1: Transition locations.

- **Comment:** Line 389 ff and Fig. 15: I'm not sure if I have fully understood your explanation., If you are changing $\omega$ (!) only by a factor of 3, tip-speed-ratio and angles of attack may vary as well that strong, so that your blade fully falls out of a meaningful operating range. On the other side transition location "only" varies by a factor less that two. Please explain in more detail.
  **Answer:** We agree that if the blade operates optimally at a rotation speed $\omega_{opt}$, which is the rotation speed in the RANS computations, then $0.5 \times \omega_{opt}$ and $1.5 \times \omega_{opt}$ are not in the (normal) operating range of the blade. We selected those $\omega$ to force the effects of the rotation speed variation on transition to be more pronounced since transition seemed not to depend strongly on $\omega$ for the studied cases. The selected rotation speeds may also occur in a transient way, such as during the accelerating phase of the wind turbine. We adopted the approach of Du and Selig [3] of keeping the angle of attack constant while changing the rotation speed. This was done to segregate the effects of the variation of the spanwise velocity as well as Coriolis and centrifugal forces from those caused by the variation of the angle of attack. For this reason, the transition locations vary less than $\omega$. These points have been made clearer in the fifth paragraph of Section 4.4 of the revised document: "The effects of rotation on the spanwise velocity are investigated using the approach of Du and Selig [3], in which the rotation speed is varied while the angle of attack is kept constant. This allows for segregating the effects of the variation of the spanwise velocity as well as Coriolis and centrifugal forces from those caused by the variation of the angle of attack."

- **Comment:** Line 392: "accelerates transition". I think "accelerate" is not the right expression. What about "shifts the transition location closer to the nose"?
  **Answer:** The authors are thankful for the suggestion and have changed the sentence in the eighth paragraph of Section 4.5 of the revised document to "The trend shown in the picture

indicates that the increase in the rotation speed shifts the transition location closer to the nose."

- **Comment:** Line 457: "reliable estimate". Again, please state accuracy of your model more quantitatively
  **Answer:** We have changed the sentence in the second paragraph of the Conclusions of the revised document in the same way as stated in the answer to "Comment: Line 14...".

**References**

[1] C. Bak et al. "Light Rotor: The 10-MW reference wind turbine". In: *Proceedings of EWEA 2012 - European Wind Energy Conference & Exhibition*. 2012.

[2] M. Drela and M. B. Giles. "Viscous-Inviscid Analysis of Transonic and Low Reynolds Number Airfoils A Theory for Predicting the Turbulent-Spot Production Rate". In: *AIAA Journal* 25.10 (1987), pp. 1347–1355. DOI: 10.2514/3.9789.

[3] Z. Du and M. S. Selig. "The effect of rotation on the boundary layer of a wind turbine blade". In: *RENEW ENERG* 20 (2000), pp. 167–181. DOI: 10.1016/S0960-1481(99)00109-3.

[4] A. van Garrel. *Integral Boundary Layer Methods for Wind Turbine Aerodynamics - A Literature Survey*. Technical report ECN-C–04-004. Energy Research Centre of the Netherlands, 2004.

[5] L. M. Mack. *Transition prediction and linear stability theory*. Technical report AGARD-CP-224. AGARD, 1977.

[6] R. E. Mayle. "A Theory for Predicting the Turbulent-Spot Production Rate". In: *Journal of Turbomachinery* 121.3 (1999), pp. 588–593. DOI: 10.1115/1.2841356.

[7] Ö. S. Özçakmak et al. "Laminar-turbulent transition characteristics of a 3-D wind turbine rotor blade based on experiments and computations". In: *Wind Energy Science* 5.4 (2020), pp. 1487–1505. DOI: 10.5194/wes-5-1487-2020. URL: https://wes.copernicus.org/articles/5/1487/2020/.

---

## Author Comment (AC2) · 21 Feb 2021

**Response to Reviewer's Comments concerning wes-2020-107**

Thales Fava          Mikaela Lokatt          Niels Sørensen
Frederik Zahle          Ardeshir Hanifi          Dan Henningson

We would like to thank the reviewer for careful and thorough reading of this paper. Our response follows.

**Referee #2 Comments**

**Comments:**

- **Comment:** 1. Introduction – The linear stability of the flow over swept wings could have been further reviewed considering that the 3D boundary layer and PSE formulation was initially made for these kind of problems
  **Answer:** The authors are thankful for the comment and have included a more extended review of the stability of the flow over swept wings in the fourth paragraph of the Introduction of the revised document.

- **Comment:** 2. Equation 29 – subscript 1 missing for the streamwise coordinate and $\gamma$ has been used both for the frequency and as intermittency factor in the RANS modelling.
  **Answer:** $\alpha$ is only a function of $x_1$ and $x'$ is a dummy variable of integration. For this reason, there is no index for it. We have changed the variable describing the frequency from $\gamma$ to $\omega$ to avoid confusion with the intermittency factor $\gamma$. We have also changed the angular velocity from $\omega$ to $\Omega$ and the wave propagation angle from $\eta$ to $\Psi$.

- **Comment:** 3. Overall, the discussion is quite thorough, however a bit prolix. For instance, a lot of effort has been placed in describing the shape of the velocity profiles but what are their implications in terms of boundary layer transition. As already reported on swept wings the highly inflectional nature of the profile of the transverse component (in this case u2), in the in-board region in figure 6(a) and 6(b) is also an indication of the potential crossflow instability. In fact, there is very little discussion on the behaviour of the mean flow and the possible route to transition, mainly focused towards the end of section 4.2.
  **Answer:** Thank you for this observation. We have included a more thorough discussion relating the velocity profiles to possible transition mechanisms in Section 4.4 (formerly 4.2).

- **Comment:** 4. Similarly, the discussion on the flow over geometry 1 and geometry 2 are completely segregated. Since the topology of the flow is quite different from each other it will be interesting to compare them right from the beginning and this will already set the scene for how the modification of the mean flow by varying different parameter will favour a particular route to transition.
  **Answer:** We have changed the text in Section 4.4 (formerly 4.2) to include a comparison between the flows in Geometries 1 and 2 and how this may favor a particular transition mechanism.

- **Comment:** 5. The sentence starting at line 349 and ending at line 350 is a bit of a contradictory statement.
  **Answer:** The sentence is "This fact means that the RANS base-flow becomes turbulent (stable) too early, before a mode could reach $N_{crit}$". The intended meaning of this sentence was that, in

the PSE analysis of the RANS base-flow, the modes do not amplify enough to reach the critical $N$-factor because the RANS transition model indicates earlier transition, rendering the velocity profiles linearly stable. We have removed this sentence as well as references to the PSE RANS approach. This was done because we believe that there are not enough converged modes in this PSE analysis to generate a reliable envelope of $N$-factors.

- **Comment:** 6. Line 353 – "These differences arise from the pressure distribution from XFOIL not exactly matching those from RANS, although they are close to each other". Any idea why there is this mismatch?
  **Answer:** A possible source of those differences is a small mismatch between the angles of attack (AoA) of XFOIL and RANS. The XFOIL computations are for an AoA calculated based on the inflow velocity and that generated by the blade rotation, which may differ from the actual AoA in the RANS simulation. Moreover, XFOIL $C_p$ distributions were obtained for a two-dimensional section of the wing, without considering its spanwise variation and the three-dimensionality of the flow present in the RANS results. Those effects are particularly important for Geometry 1 at $r_0/R = 0.26$. This explanation has been included in the first paragraph of Section 4.2 of the revised document.

- **Comment:** 7. Figures 10 and 11 can be combined and similarly figures 15 and 16
  **Answer:** We have merged figures 10 and 11 (now figure 10) and figures 15 and 16 (now figure 14) in the revised document. We have also combined figures 4 and 5 (now figure 5), figures 13 and 14 (now figure 13), and figures 18 and 19 (now figure 16).

- **Comment:** 8. Figures 12 and 17 can be rearranged so that they do not occupy a full page.
  **Answer:** We have rearranged figures 12 and 17 (now figures 11 and 15) to reduce their space.

- **Comment:** 9. The angle $\eta$ could be described in the schematic in Figure 1 for readers who are not used to the 3D flow topology, may be just sketch of the flow topology such as the development of the skin friction lines. In fact, why not show the skin friction lines from the RANS simulation also to complement some of the arguments about the three-dimensionality being more pronounced on some part of the rotor.
  **Answer:** For clarity, we have included in Figure 1 a diagram of a 3D boundary layer, describing the angle $\eta$ (now $\Psi$) between the wave propagation direction and the inviscid streamline. We agree with the referee that it would be interesting to present the skin friction lines or the streamlines over the blade surface to describe the three-dimensionality of the flow. However, in order to increase the conciseness of the article, we cited in the third paragraph of Section 4.3 two references [3, 1] that have numerically studied the flow in Geometry 2 for the same operating conditions of the present work. They present the streamlines over the blade surface and discuss three-dimensionality effects.

- **Comment:** 10. The sentence starting from line 402 could be rephrased to be more explicit.
  **Answer:** The original sentence is "The smaller sensitivity of transition to variations in the rotation speed ensues from the fact that the airfoils of Geometry 2 maintain favorable pressure gradients over a larger chordwise extent, which makes the rotation effects have smaller relative importance." We have changed it in the tenth paragraph of Section 4.5 of the revised article to: "The transition location moves less with the rotation speed for Geometry 2 because this blade maintains a non-negligible pressure gradient over a larger chordwise extent, overtaking rotation effects."

- **Comment:** 11. The splitting of the near-wall lobe of the TS eigenfunction is also observed in the presence of the large adverse streamwise pressure gradient; therefore it might be worth tying this with the strong 2D mode amplification.
  **Answer:** We are thankful for the suggestion. We have included a discussion in the sixth paragraph of Section 4.5 of the revised document linking the appearance of a second peak in the eigenfunction to the two-dimensional amplification of the mode in the presence of an adverse

pressure gradient. We have also added to Fig. 13 the modes obtained with the PSER 2D approach to show that the appearance of a near-wall peak is related to 2D TS waves.

- **Comment:** 12. The attachment line has been mentioned on quite a few occasions however there is no mention of whether it is below the threshold for contamination, keeping in mind that the leading edge radius of curvature can be quite considerable in the inboard region of the rotor.

  **Answer:** Thanks for the remark. We can define $\overline{R} = \left( \frac{u_\infty R \sin \phi \tan \phi}{2\nu} \right)^{1/2}$, where $u_\infty$ is the incoming infinite velocity, $R$ is the curvature radius of the leading edge, $\phi$ is the sweep angle, and $\nu$ is the kinematic viscosity. Contamination occurs for $\overline{R} \gtrsim 250$ [2]. In the analyzed wind-turbine blades, $\overline{R}$ is well below this threshold. The maximum values of this parameter for Geometries 1 and 2 are $\overline{R} = 41$ and $\overline{R} = 15$ at $r/R_0 = 0.26$ and $r/R_0 = 0.40$, respectively. Thus, attachment-line contamination is not expected to occur. We have included this analysis in the first paragraph of Section 4.1 of the revised document.

- **Comment:** 12. Although the method developed here will be useful in the design and optimisation of wind turbine rotor blades, the linear PSE has its limitations which D. Henningson and A. Hanifi will definitely agree, therefore it might be worth mentioning those, just to keep the reader informed and avoid any bias within the transition community.

  **Answer:** We appreciate the remark. We have stated the limitations of the PSE in predicting transition in the third paragraph of the Introduction of the revised document: "However, there are limitations in the linear PSE approach, which are the inability to predict: i) transition in strongly non-parallel flows with rapid variation in the streamwise direction; ii) transition in strong three-dimensional flows; iii) transition caused by global instability, as in the case of strong separation bubbles."

**References**

[1]  S. G. Horcas et al. "Rotor-tower interactions of DTU 10MW reference wind turbine with a non-linear harmonic method". In: *WIND ENERG* 20 (2017), pp. 619–636. DOI: `10.1002/we.2027`.

[2]  D. I. A. Poll. "Some aspects of the flow near a swept attachment line with particular reference to boundary layer transition". In: *CoA Report 7805, Cranfield University* (1978). URL: `https://dspace.lib.cranfield.ac.uk/handle/1826/832`.

[3]  F. Zahle et al. "Comprehensive Aerodynamic Analysis of a 10 MW Wind Turbine Rotor Using 3D CFD". In: *Proceedings of the 32nd ASME Wind Energy Symposium, National Harbor, Maryland, 13-17 January 2014*. 2014.

---

## Author Comment (AC3) · 21 Feb 2021

**Response to Reviewer's Comments concerning wes-2020-107**

Thales Fava      Mikaela Lokatt      Niels Sørensen

Frederik Zahle      Ardeshir Hanifi      Dan Henningson

We would like to thank the reviewer for careful and thorough reading of this paper. Our response follows.

**Referee #3 Comments**
* * *
**Comments:**

- **Comment:** L38-40: It is written that the PSE model are computational costly so not well suited for design. Nonetheless, the proposed model is based on PSE approach?
  **Answer:** We are thankful for noticing this mistake. The sentence has been corrected in the third paragraph of the Introduction of the revised document to: "The DNS approach for transition prediction provides accurate results, but it implies a high computational cost. With the current available computational power, simulations at Reynolds numbers corresponding to those on real wind turbines are not possible. The PSE analysis has a much lower computational cost compared to DNS [5], but it provides more accurate transition predictions than the RANS approach with an algebraic-integral or transport model."

- **Comment:** L50: It is mentioned that the reference RANS solver (EllipSys3D) integrate a transition prediction tool based on database method. Which one? Are there any references available? In the previous sentence, Sorensen 2009 is given as a reference but this paper deals with gamma-Retheta method which is not a database method. Additionally, since RANS results will be used as comparison for transition prediction, does the database integrates 3D effect and/or rotational effects?
  **Answer:** The referee's comment made us realize that the reference Sorensen (2009) was not the correct one to refer to the transition model used in the EllipSys3D solver for the current computations. The correct transition model is the semiempirical $e^N$ method of Drela and Giles [2, 5]. This transition model does not integrate 3D nor rotational effects. This has been changed in the fifth paragraph of the Introduction of the revised document to: "Transition prediction within this solver is obtained through the semiempirical $e^N$ method of Drela and Giles [2, 5]. This transition model does not account for effects of the blade rotation or the three-dimensional flow."

- **Comment:** L73: The equations (for the mean flow as well as for the fluctuations) are given in the general case of compressible flow including the equation for the temperature (or the energy). Is there any interest in considering the general compressible case or could it be restricted to simplified incompressible formulation? For an angular velocity of omega=1 rad/s and a radius of r=100m the azimuthal velocity at the tip will be around 100m/s ie a Mach number of 0.3 which is the usual limit to separate incompressible to compressible regimes. Additionally, the RANS code is incompressible (mentioned Section 4, p10).
  **Answer:** We have implemented our model in the bl3D and NOLOT codes, which are validated and general-purpose boundary-layer and PSE codes, respectively. These codes use a compressible formulation. We agree with the reviewer that compressibility does not play a significant role in the current cases. However, since both codes run fast and do not present issues in the incompressible limit, we thought it was unnecessary to change their formulations to an incompressible

one. Thus, the compressible equations presented in the paper reflect the original equations implemented in the bl3D and NOLOT codes.

- **Comment:** L82: cp and gamma have already been used before. cp at the end of the abstract to refer to pressure coefficient and gamma in the gamma-Retheta model (ie referring to the intermittency function). It will be used again at the bottom of page 10 to refer to the intermittency factor. Additionally, page 9, equation 29 gamma is used as the angular frequency of the disturbances. A list of symbols at the beginning of the article will be helpful.
  **Answer:** The referee's comment made us realize that the pressure coefficient should be "$C_p$" and not "$c_p$", which is the specific heat at constant pressure. In the revised article, $\gamma$ has been used only to refer to the intermittency factor, whereas $\omega$ has been used to refer to the angular frequency of the disturbances and $\Omega$ to the angular speed of the wind turbine. We agree with the reviewer that a list of symbols at the beginning would be helpful for the readers. There was such a list in a previous version of the document. However, including a list of symbols does not seem to be a common practice in WES. For this reason, we opted for removing it to comply with what seemed to be the editorial standard.

- **Comment:** L123: It is mentioned that the pressure can be obtained from the velocity and temperature variables. For me one quantity is missing: the density or the total pressure.
  **Answer:** We agree with the referee that one needs the density or the total pressure to obtain the static pressure. In the second paragraph of Section 2.1.3 of the revised article, we have added the information that we used a reference density and the temperature obtained with the BL code to compute the pressure: "The density is calculated from the temperature and pressure using the equation of state and the BL approximation of pressure being constant inside the boundary layer. "

- **Comment:** L179-180: The PSE is derived from the continuity,  momentum (better suited), energy and state equations.
  **Answer:** We have changed from "Navier-Stokes" to momentum in the first paragraph of Section 3 of the revised document.

- **Comment:** L198 (Eq 27): Even though periodicity is assumed in x2 direction, 'q' and 'qtilde' depends on x2. Same remark for eq 28.
  **Answer:** The variables 'q' and '$\tilde{q}$' in equations 27 and 28 do not depend on $x_2$. They only depend on $x_1$ and $x_3$.

- **Comment:** L205: Already mentioned but the angular frequency is quoted as gamma which has already been used as the intermittency.
  **Answer:** We have removed this ambiguity. $\gamma$ has been used only to refer to the intermittency factor, whereas $\omega$ has been employed to denote the angular frequency of the perturbations.

- **Comment:** L217 (Eq33): the density fluctuation should also tend to 0 (or better be bounded) far away from the wall ($x_3 \to \infty$)
  **Answer:** The reviewer is correct. However, since the PSE equations are differential equations of eight order, only eight boundary conditions are requested. Usually, these eight boundary conditions are imposed on the velocity and temperature perturbations. However, the far-field boundary condition for the wall-normal velocity component can be replaced with the one for the density. We have mentioned this in the fourth paragraph of Section 3 of the revised manuscript.

- **Comment:** L221: The definition of the N factor should be given here. How is it related to the amplitude function of the disturbances? Moreover it is often specified that it is an envelope N factor but never said that the envelope is a local maximum on frequency (I guess).
  **Answer:** We are thankful for the remarks. We have included this definition at the end of Section 3 of the revised manuscript:

" In the so-called $e^N$ method, transition location is predicted based on the amplification of disturbances presented by the $N$-factors computed as

$$N = \ln\left(A/A_0\right) = \int_{x_0}^{x_1} \sigma(x)dx, \tag{1}$$

where $A$ is the amplitude of the perturbations ($A_0 = A(x_0)$), $x_0$ the location where the perturbation first start to grow and $\sigma$ the growth rate of the perturbation kinetic energy $E$ defined as [3]

$$\sigma = \frac{1}{h_1}\left[-\text{Im}(\alpha) + \text{Re}\left(\frac{1}{E}\frac{\partial E}{\partial x_1}\right)\right], \tag{2}$$

$$E = \int_0^\infty \overline{\rho}\left(\hat{u}_1^2 + \hat{u}_2^2 + \hat{u}_3^2\right)dx_3." \tag{3}$$

In the present work, we use the so-called *envelope-of-envelopes* method [1]. This means that transition is predicted by the envelope of $N$-factors, with all curves being computed for different pairs of $\omega$ and $\beta$. This information has also been included at the end of Section 3.

- **Comment:** L255: It is mentioned that cp distributions are approximation/differ from RANS ones. It would be interesting to illustrate (quantify) this discrepancy on a figure all the more than in the following this argument is reiterated to justify the differences between BLX and RANS velocity profiles L307 as well as on transition locations L353 and L357.
  **Answer:** We are thankful for the suggestion. We have added a section in the revised document (Section 4.2) to include Fig. 1 and discuss the differences between RANS and XFOIL pressure distributions.

[Figure]

Figure 1: Comparison between XFOIL and RANS pressure distributions for the suction side of the airfoils of Geometries 1 and 2 at three radial positions.

- **Comment:** L284: "Although exhibiting higher values, the BLR and BLX profiles of spanwise velocity present the same shape of those from RANS." I do not understand this sentence since in Fig 6b, RANS provide a u2 profile lower than zero (except close to the wall) while the u2 profile provided by BLR and BLX is positive.
  **Answer:** We agree with the reviewer that this sentence is not correct. We have corrected it in the second paragraph of Section 4.4 of the revised document to: "Although the BLR and BLX profiles of spanwise velocity are close to each other, they indicate a positive velocity (flow towards the tip of the blade) whereas the spanwise velocity profile from RANS is only positive in the near-wall region."

- **Comment:** L294: "This is similar to what was observed in Geometry 1 and may indicate the three-dimensional character of the flow at lower radii". Not agree, related to the previous remark.
  **Answer:** We have changed the sentence in the fifth paragraph of the revised document to improve its accuracy: "This also occurs in a smaller extent at the inner radial position of Geometry 1 (Figs. 6a and 6b) where, at the near-wall region, the spanwise velocity profile presents an inversion of direction. The fact that the inversion of the spanwise velocity profile only occurs at the inner radial position of Geometries 1 and 2 may confirm the three-dimensional character of the flow at lower radii."

- **Comment:** L337: The threshold value for eN method has been set up to 9 which is a typical value considering flight tests and quiet wind tunnel tests. Is this value appropriated for wind turbine blade application in the 'atmospheric boundary layer'?
  **Answer:** We agree with the reviewer that $N = 9$ represents transition in an environment with very low turbulence intensity (0.07 % according to Mack's relation [4]), not representative of all atmospheric conditions. This value was selected in order to have a larger region of laminar flow in the RANS results, allowing a more detailed comparison between the results from the developed model and RANS, both in terms of the boundary-layer profiles and the transition location. The following sentence has been added to the first paragraph of Section 4.5 of the revised manuscript: "Although not representative of all atmospheric conditions, it is assumed $N_{crit} = 9$ in the current work to have a larger region of laminar flow in the RANS results, allowing a more detailed comparison between the developed model and RANS."

- **Comment:** L348: The PSE RANS results (not shown). It is unfortunate, since these results are the one to be considered as the reference to validate your approach. Is it possible to perform laminar RANS computations switching off the database transition prediction? Or rising the value of the transition threshold to obtain a laminar extend as long as possible in the RANS computation in order to analyze its stability with PSE? If not at least compare the evolution of the N factor between RANS and BLX for the first per cent of chord.
  **Answer:** We used a value of critical $N$-factor corresponding to natural transition ($N = 9$) to force a larger laminar region in the RANS computations. The PSE analysis of the RANS base-flow revealed some modes amplified to a value lower than the critical $N$-factor. However, in most cases, spurious modes with large growth rates were obtained. The latter is probably related to numerical errors in the base-flow derivatives that had to be computed in the post-processing step. There were oscillations in some high-order derivatives that may stem from the precision of the base-flow variables as output by the EllipSys3D RANS solver and/or spatial oscillations in the flow field. Thus we have opted for removing references to the PSE RANS approach since we believe that there are not enough converged modes in this PSE analysis to generate a reliable envelope of $N$-factors.

- **Comment:** L384: "However, the modes tend to have a single-peaked structure at $r_0/R = 0.26$, associated with their high propagation angle (in absolute value)." This has to be related to the spanwise mean velocity profile (fig 6a and b) which is inflectional (ie sensitive to CF like disturbances)
  **Answer:** We agree with the reviewer that the single peak in the eigenfunction at $r_0/R = 0.26$ is related to the inflectional spanwise velocity profile, which indicates sensitivity to crossflow-related disturbances. We have added the following explanation in the sixth paragraph of Section 4.5 of the revised manuscript: "The modes tend to have a single peak at this location, associated with their high $|\Psi|$ and the inflectional spanwise velocity (Fig. 6b). This indicates that transition may be triggered by oblique TS or crossflow modes."

- **Comment:** L434: "The single-peaked modes observed at $r_0/R = 0.26$ for Geometry 1 and $r_0/R = 0.40$ for Geometry 2 ($\omega = 0.45$ rad.s-1) might represent an intermediate stage between a TS and crossflow transition". Is there a shift/reduction of the frequency of the instability responsible for transition onset (the one reaching the Ncrit).
  **Answer:** The reduced frequency F of the perturbations leading to the transition onset in Geometries 1 and 2 are presented in Table 1. F decreases with the rotation speed (not monotonically

in Geometry 1 at $r_0/R = 0.26$ and Geometry 2 at $r_0/R = 0.89$). For low $\Omega$, F is smaller at the inner radial position ($r_0/R = 0.26$ and $r_0/R = 0.40$). The increase in $\Omega$ makes F at these locations become larger than at higher radii.

| Geometry 1 | | |
|---|---|---|
| | $r_0/R = 0.26$ | $r_0/R = 0.58$ | $r_0/R = 0.89$ |
| $\Omega = 0.32$ rad.$s^{-1}$ | F=1.86956E-05 | F=1.98738E-05 | F=2.19379E-05 |
| $\Omega = 0.64$ rad.$s^{-1}$ | F=1.98087E-05 | F=1.60444E-05 | F=1.60297E-05 |
| $\Omega = 0.96$ rad.$s^{-1}$ | F=1.54694E-05 | F=1.32022E-05 | F=1.43235E-05 |
| Geometry 2 | | |
| | $r_0/R = 0.40$ | $r_0/R = 0.58$ | $r_0/R = 0.89$ |
| $\Omega = 0.45$ rad.$s^{-1}$ | F=1.77893E-05 | F=1.88131E-05 | F=2.68559E-05 |
| $\Omega = 0.90$ rad.$s^{-1}$ | F=1.64016E-05 | F=1.29000E-05 | F=1.15029E-05 |
| $\Omega = 1.35$ rad.$s^{-1}$ | F=1.25585E-05 | F=7.95290E-06 | F=1.21288E-05 |

Table 1: Reduced frequency of the perturbations leading to the transition onset.

**References**

[1] D. Arnal and G. Casalis. "Laminar-turbulent transition prediction in three-dimensional flows". In: *PROG AEROSP SCI* 36 (2000), pp. 173–191. DOI: 10.1016/S0376-0421(00)00002-6.

[2] M. Drela and M. B. Giles. "Viscous-Inviscid Analysis of Transonic and Low Reynolds Number Airfoils". In: *AIAA Journal* 25.10 (1987), pp. 1347–1355. DOI: 10.2514/3.9789.

[3] A. Hanifi et al. *Linear nonlocal Instability Analysis - the linear NOLOT code*. Technical report FFA-TN 1994-54. Bromma, Sweden: The Aeronautical Research Institute of Sweden, 1994.

[4] L. M. Mack. *Transition prediction and linear stability theory*. Technical report AGARD-CP-224. AGARD, 1977.

[5] Ö. S. Özçakmak et al. "Laminar-turbulent transition characteristics of a 3-D wind turbine rotor blade based on experiments and computations". In: *Wind Energy Science* 5.4 (2020), pp. 1487–1505. DOI: 10.5194/wes-5-1487-2020. URL: https://wes.copernicus.org/articles/5/1487/2020/.